# Space Station conditions are selective but do not alter microbial characteristics relevant to human health

Maximilian Mora[1], Lisa Wink[1], Ines Kögler[1], Alexander Mahnert[1], Petra Rettberg [2], Petra Schwendner[3], René Demets[4], Charles Cockell [3], Tatiana Alekhova[5], Andreas Klingl[6], Robert Krause[1,7], Anna Zolotariof [3], Alina Alexandrova[5] & Christine Moissl-Eichinger[1,7]

The International Space Station (ISS) is a unique habitat for humans and microorganisms. Here, we report the results of the ISS experiment EXTREMOPHILES, including the analysis of microbial communities from several areas aboard at three time points. We assess microbial diversity, distribution, functional capacity and resistance profile using a combination of cultivation-independent analyses (amplicon and shot-gun sequencing) and cultivation-dependent analyses (physiological and genetic characterization of microbial isolates, antibiotic resistance tests, co-incubation experiments). We show that the ISS microbial communities are highly similar to those present in ground-based confined indoor environments and are subject to fluctuations, although a core microbiome persists over time and locations. The genomic and physiological features selected by ISS conditions do not appear to be directly relevant to human health, although adaptations towards biofilm formation and surface interactions were observed. Our results do not raise direct reason for concern with respect to crew health, but indicate a potential threat towards material integrity in moist areas.

[1] Medical University of Graz, Department of Internal Medicine, Auenbruggerplatz 15, 8036 Graz, Austria. [2] German Aerospace Center (DLR), Institute of Aerospace Medicine, Radiation Biology Department, Research Group Astrobiology, Linder Höhe, 51147 Cologne, Germany. [3] University of Edinburgh, School of Physics and Astronomy, James Clerk Maxwell Building, Peter Guthrie Tait Road, Edinburgh EH9 3FD, UK. [4] European Space Research and Technology Centre (ESTEC), Keplerlaan 1, 2201 AZ Noordwijk, The Netherlands. [5] Lomonosov Moscow State University, Biological Faculty, ul. Leninskiye Gory, 1, стр. 12, Moscow, Russia. [6] Ludwig Maximilians University of Munich, Plant Development and Electron Microscopy, Department of Biology I, Biocenter, Großhaderner Str. 2, 82152 Planegg-Martinsried, Germany. [7] BioTechMed Graz, Mozartgasse 12/II, 8010 Graz, Austria. Correspondence and requests for materials should be addressed to C.M.-E. (email: christine.moissl-eichinger@medunigraz.at)

Human space exploration beyond boundaries of Earth and Moon is a declared goal of NASA, ESA, Roscosmos and other space-faring agencies, envisaging a potential human Mars mission in the next 20–30 years. Maintenance of crew's health during a several hundred days journey in a confined artificial environment in space is one of the key aspects, which has to be addressed.

The human immune system was shown to be compromised under space flight conditions, as a significant decrease of lymphocytes and also of the activity of innate and adaptive immune response was observed[1,2]. Adding an order of complexity, human health is intertwined with its microbiome, billions of microorganisms thriving on external and internal surfaces of the human body. Our body's microbiome is prone to external factors, including the environmental microbiome, as they are in constant exchange and interaction.

Several risks with respect to microorganisms and human spaceflight have been identified. These include a potentially increased infection risk, as it has been shown that microgravity affects the virulence of certain microorganisms, such as *Salmonella typhimurium*[3], *Listeria monocytogenes*, and *Enterococcus faecalis*[4]. Another stressor for the indigenous microorganisms is the strict maintenance regime, which could result in an increase of antimicrobial resistances, as recently shown for highly-maintained, confined built environments[5]. Some microorganisms might even pose a risk to the material integrity of a spacecraft: So-called technophilic microorganisms, in particular fungi, are able to corrode alloys and polymers used in spacecraft assembly[6]. Technophilic microorganisms caused major problems on the former Russian space station MIR, partaking in damage to structural materials as well as malfunctioning of various space systems and equipment[7,8]. Specifically, *Bacillus*, *Penicillium*, and *Aspergillus* species were associated with the progressive destruction of a window in MIRs descent module[9], and mold on wiring connectors was associated with electrical outages[10].

The majority of information with respect to environmental microbiome composition and dynamics aboard manned spacecraft is retrieved from ground-based simulation studies, such as the Mars500[11] and the HI-SEAS (http://hi-seas.org/) experiments. However, the International Space Station (ISS) has, like no other currently available testbed for long-term manned space missions, the scientific benefit of providing real spaceflight conditions, including microgravity and an elevated background radiation. The ISS orbits Earth ~400 km above ground and is meanwhile constantly inhabited for more than 18 years. Except for cargo exchange and the arrival of new crew members roughly every 6 months, the ISS is completely sealed off from any biological ecosystem[12].

Analyses of the ISS microbiome have already been performed, including microbial analyses of ISS debris and dust[13–16], the study of the astronauts' microbiome[17], the characterization of bacterial and fungal isolates from the ISS[18,19], and the (molecular) microbial analysis of swab and wipe samples taken inside the ISS[20]. A study investigating the growth behavior of nonpathogenic (terrestrial) bacteria aboard the ISS found no changes in most bacteria, given that they have enough nutrients[21]. Other publications focused on the detection of antimicrobial resistance genes aboard the ISS and evaluated the potential risk these genes might represent in a closed spacecraft environment[22]. Singh et al. assessed the succession and persistence of microbial communities and the associated antimicrobial resistance and virulence properties based on metagenomic reads obtained from samples of three flights. In this study, overall 46 microbial species were found, including eight biorisk group 2 species, to be persistent on the ISS over a timespan of roughly one and a half years[23]. The authors inferred an increase of antimicrobial resistance and virulence genes over time.

Although an increased risk for the health of the astronauts and cosmonauts aboard has often been proposed, infections of crew members or health issues related to pathogenic action of microorganisms have been reported only rarely[24]. This may be attributable to environmental contamination limits (air and surfaces) having rarely been exceeded, at which times effective countermeasures had been implemented[25]. Moreover, a genomics-based meta-analysis demonstrated that although pangenomes of *Bacillus* and *Staphylococcus* isolated from the ISS differed from Earth-based counterparts, these differences did not appear to be health-threatening[26]. Thus, a comprehensive understanding of potential microbial adaptations to the ISS based on complementary genomic and cultivation approaches is important to better understand potential human health implications.

In this study, we report on the realization of the ISS experiment EXTREMOPHILES. Our objectives included the analysis of the profile, diversity, dynamics, and functional capacity of the cultivable and noncultivable microbiome aboard. Moreover, we assessed their genomic and physiological adaptation toward ISS conditions and refuted the hypothesis, that, as indicated by previous literature reports, ISS microorganisms possess a higher extremo-tolerance and antibiotics-resistance potential compared with ground controls. Moreover, we observed surface-microbe interaction with regard to material integrity, exhibited by selected, freshly isolated ISS strains.

## Results

**Study set-up and sampling overview.** In-flight sampling on board the ISS was performed April to June 2017 under the ESA operation named EXTREMOPHILES. All samples ($n = 24$, plus controls) were taken during three sampling sessions. With a span of 72 days between session A and B they were conceptualized for time-course sampling (same sampling locations). For comparative purposes, an ISS-relevant clean room and a therein housed cargo-spacecraft were sampled, namely clean room S5C at the Centre Spatial Guyanais near Kourou in French Guiana with an ATV spacecraft.

**ISS microbiome is dominated by human-associated microorganisms.** The microbial community composition was assessed by amplicon sequence analysis of wipe samples obtained from sessions A–C and the Kourou clean room. Archaeal and bacterial RSVs (>3.500) were retrieved from ISS samples with the universal approach (Fig. 1). The signatures belonged to 377 genera, with *Streptococcus*, *Corynebacterium*, *Lactobacillus*, *Acinetobacter*, *Staphylococcus* as dominant taxa (Fig. 1, Supplementary Fig. 1, Supplementary Data 1). Firmicutes, Proteobacteria, Actinobacteria, and Bacteroidetes were found to be the predominant bacterial phyla (all samples), whereas archaeal signatures (Woesearchaeota, Thaumarchaeota, and Euryarchaeota) were frequently detected in air (Cupola air: 14.1% and Columbus air session B: 3.6% of all sequences) and on various surfaces (Fig. 1). By the Archaea-targeting approach, mostly gut associated *Methanobrevibacter* sequences (surface Cupola, Waste and Hygiene Compartment), Woesearchaeota (ATU; hand grips Columbus), and unclassified Archaea (ATU, dining table) were detected. To a lesser extent, we also found signatures from Halobacteria (SSC Laptop Columbus) and Thaumarchaeota (dining table). A phylogenetic tree displaying the archaeal diversity is given in Supplementary Fig. 2.

The clean room samples, which were analyzed for comparison, showed a different microbial signature profile, with a predominance of alpha-proteobacterial genera (*Sphingomonas*, *Novosphingobium*, and *Methylobacterium*).

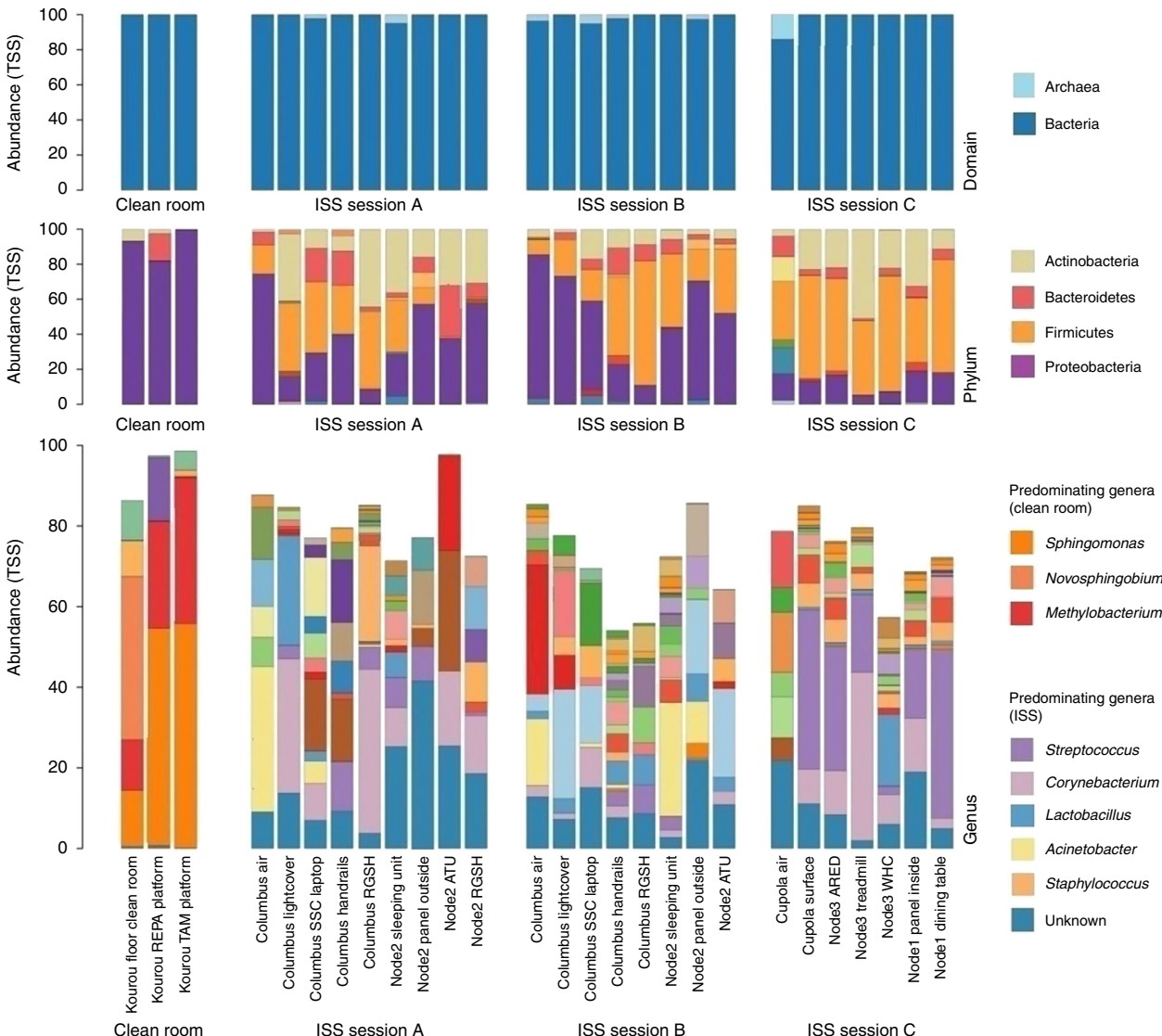

**Fig. 1** Microbiome composition in clean rooms (left column) and the ISS. Samples were taken during session A, B, and C. A full version (with additional details) of the figure is available in the Supplementary Fig. 1

**The microbiome aboard the ISS changes over time**. To retrieve insights into the microbiome changes over time, sessions A and B were conceptualized with a time-lapse of 72 days between the two samplings. During the complete sampling timeframe, no crew exchange took place, but two cargo deliveries docked (SpaceX and Soyuz). Notably, the microbial diversity (RSVs) was found to be increased significantly in samples of session B ($p = 0.034$, ANOVA; Inverse Simpson's; Supplementary Fig. 3a), however, the evenness of samples did not change significantly ($p = 0.68$, ANOVA). ANOSIM analysis indicated a significantly different composition of the samples taken in session A and B ($p = 0.001$). LEfSe analysis (targeting the 300 most abundant genera) identified a substantial increase in signatures belonging to typically gastrointestinal tract-associated genera *Escherichia/Shigella* ($p = 0.017$, ANOVA), *Lachnoclostridium*, *Ruminococcus_2* ($p = 0.046$), and *Pseudobutyrivibrio* toward session B, whereas members of *Cloacibacterium* ($p = 0.027$) and unclassified Corynebacteriaceae ($p = 0.02$) were significantly reduced.

Significant changes on RSV level are given in Supplementary Fig. 3b, with an overall notable increase of a certain RSV of *Ralstonia* in samples of session B. Pie charts were created from single locations within the ISS to visualize the changes in

microbiome composition (Supplementary Fig. 3c, d). Signatures of unclassified Woesearchaeota (DHVEG6) were found among the 40 most abundant microbial genera (additional details below).

**The ISS harbors a core microbiome of 55 microbial genera**. Core microbiome analyses (100 most abundant RSVs), identified 34 taxa shared amongst all sampling time points (A–C, minimal relative abundance: 10%), whereas 55 taxa were shared at genus level. The most abundant, shared RSVs belonged to the microbial genera *Haemophilus*, *Gemella*, *Streptococcus*, *Corynebacterium*, *Staphylococcus*, *Lactococcus*, *Neisseria*, and *Finegoldia*. Thirty-one taxa were shared amongst all modules. To obtain a better overview on the biogeography of the ISS microbiome, a network was calculated (Fig. 2).

The network analysis showed a higher abundance for RSVs which belong to the core ISS microbiome (e.g., *Streptococcus*, *Corynebacterium*, *Staphyloccocus*, *Haemophilus*, *Gemella*, or *Propionibacterium*).

Most location-specific RSVs were observed for WHC (Waste and Hygiene Compartment) and RGSH (Return Grid Sensor Housing). This was expected for the WHC area (as hygienic activities shed (internal) human microorganisms into the

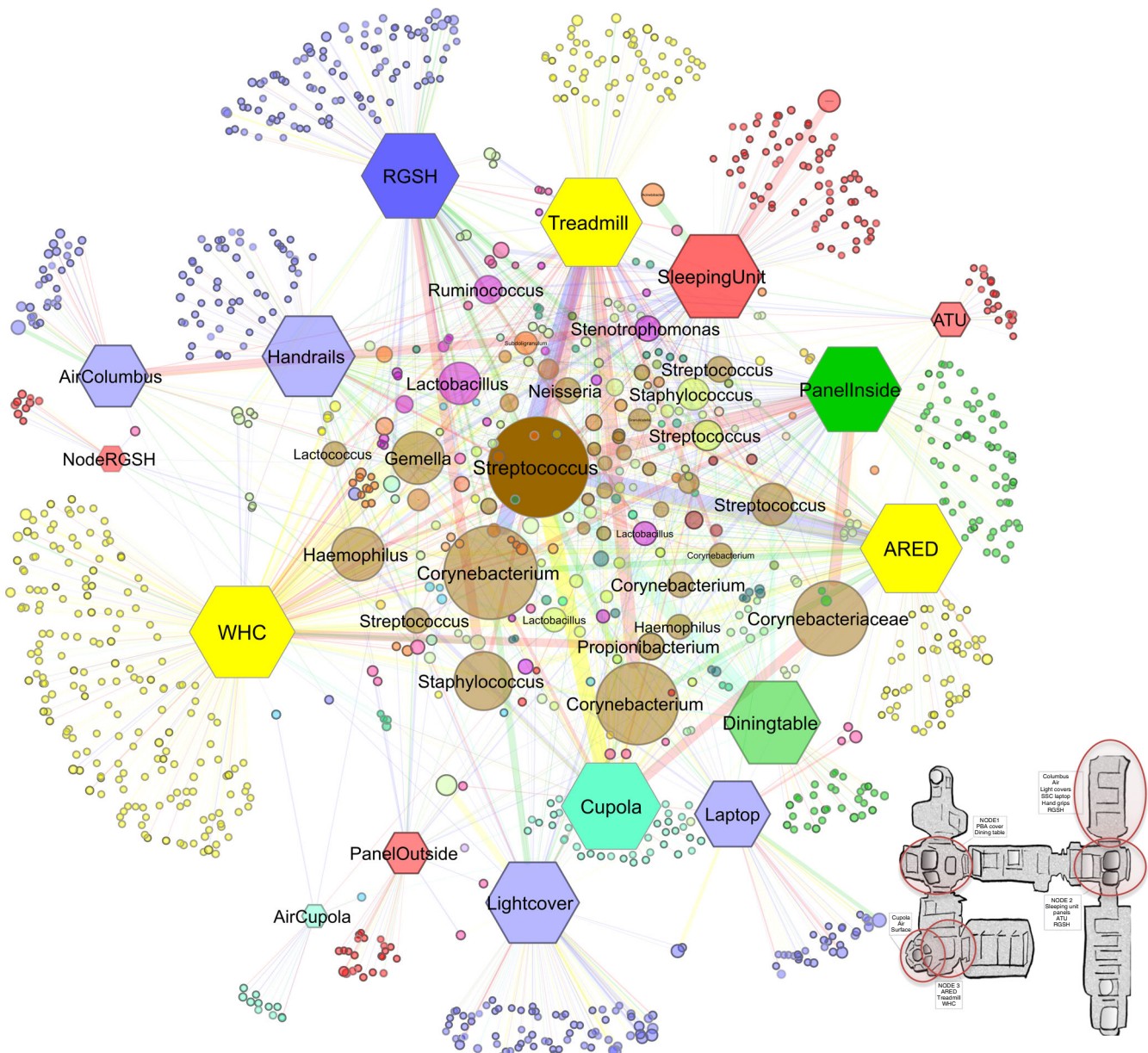

**Fig. 2** Network analysis and model of the ISS with sampling sites (indicated by red circles; full information of ISS model is given in Supplementary Fig. 4). Samples are shown as hexagons, RSVs are shown as circles. Colors of sample nodes refer to respective ISS modules. RSV nodes show mixtures of these colors if they were shared by several locations and modules of the ISS. The borders of the circles are darker, when found in several sessions. The size, transparency and the size of the node descriptions corresponds to their abundance. The thickness and transparency of the edges (lines) follows the calculated e-weights. Layout: Spring Embedded, with e-weights. color code: yellow (NODE3), red (NODE2), blue (Columbus), aqua (Cupola), green (NODE1). Abbreviations: WHC (Waste and Hygiene Compartment), ARED (Advanced Resistive Exercise Device), ATU (Audio Terminal Unit), RGSH (Return Grid Sensor Housing). A close-up model of the ISS is shown in Supplementary Fig. 4

environment), but it was surprising for the RGSH. The RGSH is the air inlet part of the air recycling system, and was therefore expected to accumulate biological burden from the environment, but not to host indigenous microbiology.

Locations with regular crew activity (e.g., treadmill, sleeping unit, handrails) showed higher proportions of RSVs assigned to the human-associated genera *Stenotrophomonas*, *Ruminococcus* and *Lactobacillus*. When clustering the samples according to their origin, the network also indicated that locations exposed to high human traffic from different modules are more similar to each other than samples of high and low human traffic which were taken within the same ISS module.

**Local factors shape microbiome composition.** We were interested in external parameters influencing the ISS microbiome. Redundancy analysis indicated a significant effect of the time of sampling (sessions; $p = 0.010$), and indicated a potential effect of the location within the ISS (module; $p = 0.054$) on microbiome composition. We further categorized the different samples into: air, personal area (sleeping unit), shared areas which are highly frequented (e.g., communication items, handrails), and shared areas which are less often touched (e.g., lightcover, RGSH, etc.). A PCoA plot performed on RSV level did not indicate a different composition of the microbiome according to these categories (Fig. 3a) ($p = 0.364$, ANOSIM based on Bray–Curtis distance

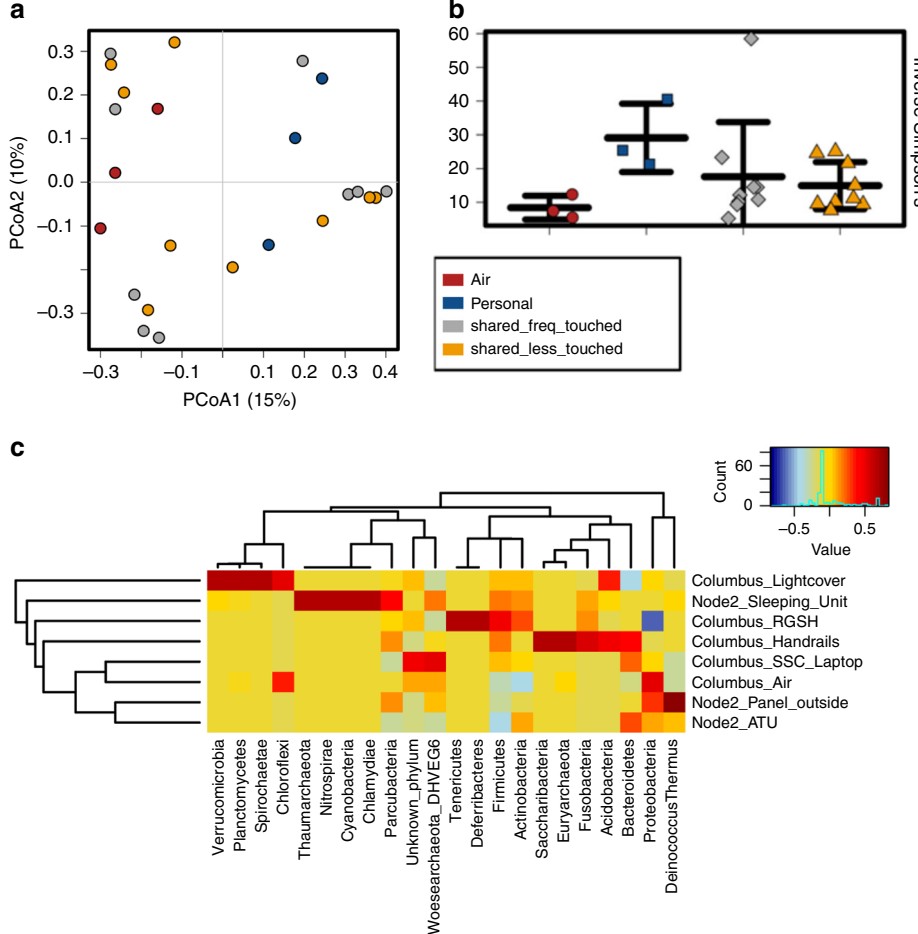

**Fig. 3** Microbiome composition according to sample categories and sample types. **a** PCoA plot of different location categories. **b** The highest diversity of microbial signatures was observed in microbiome samples from personal areas. The lowest diversity was detected in air samples ($p = 0.19$, ANOVA; sample depth rarefied to 649 reads). Error bars visualize standard deviation. **c** Hierarchical cluster analysis (Pearson's correlation; universal microbiome data set). Certain microbial phyla (in particular archaeal phyla) were found to correlate with specific sampling sites

metric). The highest diversity of microbial signatures was detected in personal areas of the ISS, without being significantly different to other area categories (Fig. 3b). Besides the results shown in Fig. 3b, the diversity analyses were performed at multiple sampling depths, and results were consistent (data not shown).

ANOVA plot analysis indicated e.g., the increased presence of human-associated *Streptococcus* RSVs in samples from dining table and workout area, and *Neisseria* species (human mucosa-associated microorganisms) were particularly detected in the sleeping unit and workout area. *Lactococcus* signatures were particularly found in samples from the dining table (potentially food-associated), the sleeping unit and the workout area, whereas RSVs from *Actinomyces*, *Enterococcus*, *Lautropia* and *Brevibacterium* were significantly enriched on handrails, in air, on the dining table, and in the workout area, respectively (all $p$-values < 0.05, Supplementary Fig. 5). The waste and hygiene area showed significantly increased abundances of RSVs belonging to *Lactobacillus*, *Propionibacterium*, *Collinsella*, *Subdoligranulum*, *Romboutsia*, and *Anaerostipes* (Supplementary Fig. 5). Although some environmental genera were also detected (e.g., *Bacillus*, *Pseudomonas*), the majority of microorganisms detected by the sequencing-based analysis appears to be human-derived across all samples. Influence of other parameters, such as local changes in humidity, different surface materials or different cleaning frequencies could not be assessed due to limitations in provided information.

According to a hierarchical cluster analysis based on Pearson's correlation across session A and B (Fig. 3c), a positive correlation of certain microbial phyla with sampled locations was found, being in agreement with findings from the cultivation assays (e.g., *Deinococcus* sp. was isolated from Node2_Panel_Outside). Columbus handrails were found to be correlated with e.g., *Saccharibacteria* and *Bacteroidetes*. A particular pattern was detected for the archaeal signatures retrieved, which were found to be indicative for the sleeping unit (Thaumarchaeota), the handrails (Euryarchaeota) and the samples from the Columbus_SSC_Laptop (Woesearchaeota).

**Functions inferred from 16S rRNA gene and metagenomic information.** As metagenomics could only be performed on pooled subset of samples (due to low biomass restrictions and sampling set-up, see below), Tax4fun analysis was initially used to predict potential microbial metabolic capabilities, their location specificity, and potential shifts over time. The LEfSe analysis result is provided in Supplementary Fig. S6, and revealed a location specific predicted capacity of the microbial community, with e.g., increased predicted functions in KEGG pathway "Base_excision_repair" in samples from the cupola module.

On gene level, different functions were predicted, indicative of a respective module. Node 3 (ARED, treadmill, and WHC) revealed predicted signatures of cobalt/nickel and antibiotic

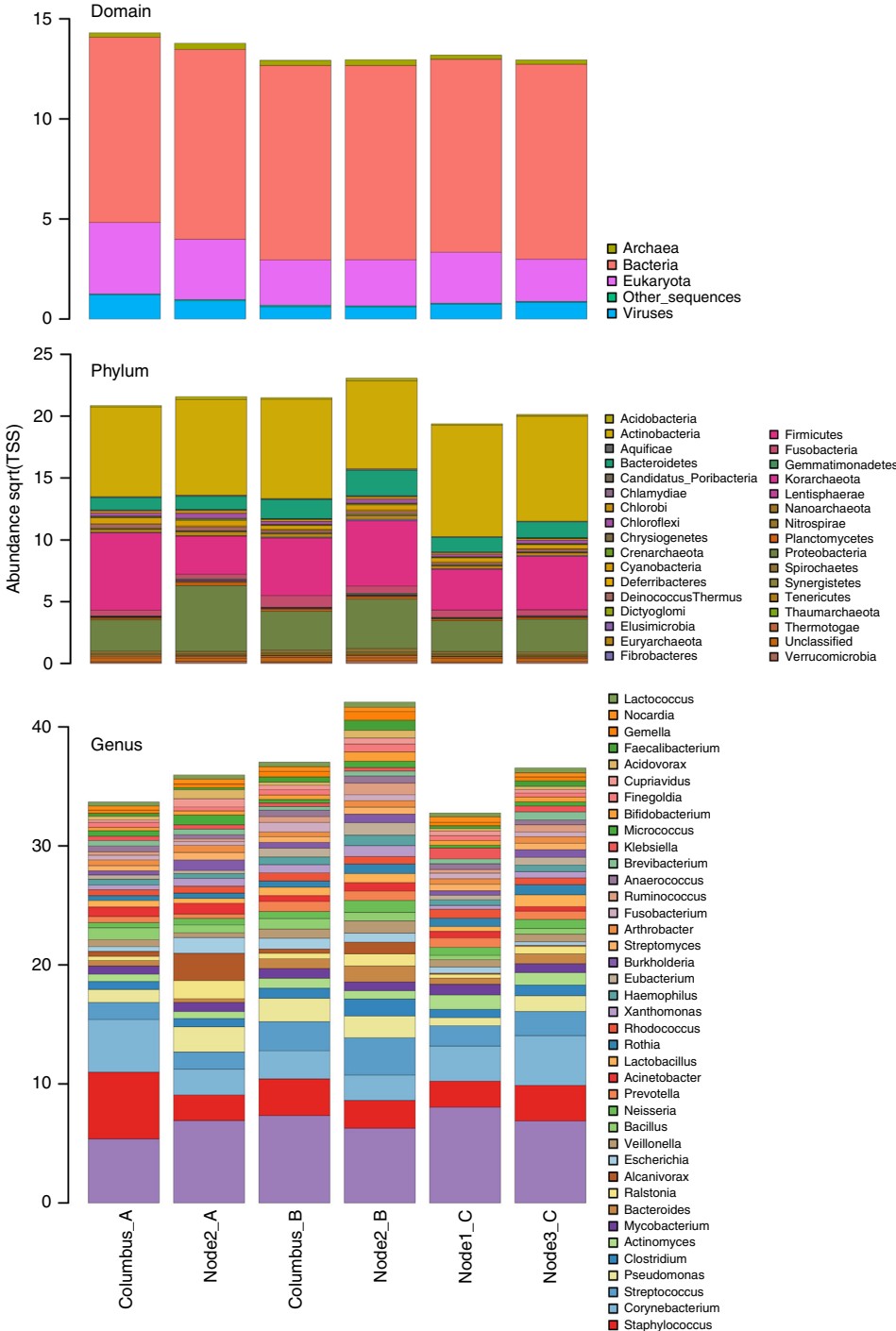

**Fig. 4** Shotgun metagenome-based taxonomic information from ISS locations. Information is given at domain, phylum and genus (top 40) level. Columbus_A refers to metagenomics information retrieved from Columbus module, session A

transport system ATP binding proteins. Notably, a specific increase of a cobalt/zinc/cadmium resistance protein was predicted for Node 2 (sleeping area and panel samples), an iron complex transport system substrate binding protein for Node 1 (e.g., dining area), and an antibiotic transport system permease protein for the cupola area. This indicated a potential microbial competition and a relevance of transition metal components (e.g., from ISS materials) for the microbial community in these environments.

For shotgun metagenomics, samples were pooled according to session and location (module). The taxonomic composition as retrieved from metagenomics was found to be somewhat different to the amplicon-based analysis. In particular the predominance of *Propionibacterium* reads was striking, as these signatures were not well reflected in the amplicon approach (Fig. 4; Supplementary Data 2).

However, *Staphylococcus, Corynebacterium, Streptococcus* could be confirmed, also in the shotgun metagenomics data set, as omnipresent on all sampled surfaces of the ISS. Core microbial taxa and functions showed a stable distribution over different fractions of samples. Hence, 57% of all taxa and 34% of all functions were shared in all samples (even higher proportions

were visible when samples were grouped per module 68% and 51% or per sampling session 73% and 58% respectively). Fungal (e.g., *Malassezia*), viral (e.g., *Microvirus*), and archaeal sequences (e.g., *Methanobrevibacter*) also belong to the core taxa of the ISS.

Sample Node2_B showed the highest Shannon diversity on genus (H' ~4) and functional level (H' ~10; Supplementary Fig. 7). Similarity estimates based on Bray-Curtis distances revealed that microbes and their functions from the Columbus module showed the biggest difference to samples from Node 1, 2, and 3. In addition the Columbus module also experienced the biggest shift of its microbiome along PCoA Axis 1 (taxa: ~60%, functions ~40%) between the first and last sampling session.

Regarding functions, genes assigned to arginine biosynthesis (amino acid metabolism), degradation of L-Ornithine (amino acid metabolism), copper-translocating P-type ATPases (virulence, disease, and defense), and the phage major capsid protein (phages, prophages, and transposable elements) were ubiquitously distributed, whereas functions assigned to dormancy and sporulation, photosynthesis, motility, and chemotaxis as well as aromatic compounds metabolism showed location-dependent variations (Supplementary Fig. 8; Supplementary Data 3). Functions involved in iron acquisition and metabolism (ferrous iron transport protein B: 0.2% in functional core), potassium metabolism, nickel ABC transporters and others were highly abundant, indicating a potential surface interaction with ISS materials.

**Cultivable microbial community reveals extremo-tolerant traits**. In the course of this study, hundreds of colonies/cultures were processed, resulting in 76 unique bacterial isolates (Supplementary Data 4). Along with the bacteria, also fungi (all biosafety risk group S1) were isolated, but were not further analyzed herein. These included *Aspergillus* species (*A. sydowii*, *A. unguis*), *Chaetomium globosum*, *Penicillium* species (*P. aurantiogriseum*, *P. brevicompactum*, *P. chrysogenum*, *P. crustosum*, and *P. expansum*), *Rhizopus stolonifera* and *Rhodotorula mucilaginosa* (see Supplementary Table 1). *P. brevicompactum*, *P. chrysogenum*, *P. crustosum*, *R. stolonifera*, and *R. mucilaginosa* may cause allergenic reactions and *R. mucilaginosa* can also act as opportunistic pathogen. Archaea could not be grown from any sampling site.

Most of the bacterial isolates showed a distinct pattern in origin distribution and special growth/enrichment characteristics (Fig. 5).

A number of isolates was obtained under stringent cultivation or pretreatment conditions. This included (i) UV- and X-ray resistant microorganisms, such as *Deinococcus marmoris*, *Curtobacterium flaccumfaciens*, *Brevibacillus agri* (UV$_{254 nm}$: 200 J/m$^2$), *Roseomonas* species, *Kocuria palustris*, *Micrococcus yunnanensis*, *Paenibacillus* sp. (X-ray: 1000 Gy), (ii) microorganisms growing particularly at high or low pH, or (iii) heat-shock survivors (*Pseudomonas psychrotolerans*, *Paenibacillus* sp.) (Fig. 5). Isolates retrieved under non-mesophile conditions were, for example, *Thermaerobacter literalis* (a true thermophile isolated at 65 °C from the ATU in Node2, no growth below 50 °C), *Sphingomonas aerolata* and *Microbacterium lemovicicum* (exhibiting extraordinary cryotolerance, isolated only at 4 °C, maximal growth temperatures were 51 °C and 32 °C, respectively).

**Microbial resistance potential is similar to that of ground controls**. We analyzed physiological characteristics of ISS microbial isolates. In particular, we were interested whether they withstand physical and chemical stressors better than same or closely related microbial species from ground controls.

For these tests, we selected a subset of microbial isolates from the ISS, spanning 11 microbial genera (listed in Fig. 6). This list included typical confined-indoor bacteria, like *Bacillus*, *Micrococcus*, and *Staphylococcus*, but also microorganisms of special interest (associated to spacecraft assembly, extraordinary hardy, extremo-tolerant) were included (e.g., *Microbacterium*, *Cupriavidus*, or *Ralstonia*). For comparative reasons, we included also microbial isolates from ground controls (clean rooms) or culture collections. Overall, the final list comprised 29 different microbial strains.

All these strains were tested with respect to heat-shock resistance in the stationary phase (*Bacillus* cultures contained spores), upper and lower temperature limit (growth), upper and lower pH limit (growth), and resistance toward a variety of antibiotics (Fig. 6). A subset of isolates was additionally analyzed with respect to tolerance against NaCl and MgSO$_4$ (details given in Supplementary Table 2).

Antimicrobial susceptibility testing for 17 clinically relevant antibiotics was performed. Antibiotic resistance/susceptibility was found to be in some cases strain-specific but mostly species/genus-specific, independent from their isolation source (ISS or ground control). In particular the tested *Bradyrhizobium* species showed a vast resistance against numerous antibiotics, as did one *Roseomonas* strain. The antibiotic resistance pattern was judged following the EUCAST guidelines for PK/PD (non-species related) or, for *Staphylococcus* isolates, *Staphylococcus* spp. breakpoints[15] (for details see the "Methods" section).

It has to be stressed that all tested isolates were nonpathogenic and that these results shall not be used for clinical risk assessment of any kind. The ISS strains were not found to be significantly more resistant (number of antibiotics or concentration) than their counterparts from ground controls.

Notably, the strains showed a growth temperature span of 18–52 °C (14–32 °C, *Roseomonas* C63; *Bacillus pumilus*, 4–56 °C). The minimal and maximal growth temperatures, or the temperature span, were not significantly different in ISS isolates compared with control microorganisms (Mann–Whitney *U* test; *p* = 0.515 (minimal temperature), *p* = 0.916 (maximal temperature), and 0 = 0.754 (temperature span)). For growth at different pH values, the isolates revealed a pH span of 4–10 pH values (*Bradyrhizobium erythroplei* LMG28425, pH3-7; *Bacillus altitudinis* R10_C4_IIIB, pH 2–12). As seen for the temperature, no significant difference in pH preference of ISS strains versus ground control strains was observed (Mann–Whitney *U* test; *p* = 0.884 (minimal pH), *p* = 0.633 (maximal pH), and *p* = 0.488 (pH span)).

**Genomic inventory of ISS microbes is similar to that of non-ISS relatives**. In order to understand the specific genomic characteristics of ISS microorganisms, we selected six different species for genome sequencing and reconstruction, namely: *Bacillus pumilus* strain pH7_R2F_2_A, *Bacillus safensis* strain pH9_R2_5_I_C. *Bradyrhizobium viridifuturi* strain pH5_R2_1_I_B, *Cupriavidus metallidurans* strain pH5_R2_1_II_A, *Methylobacterium tardum* strain pH5_R2_1_I_A and *Paenibacillus campinasensis* strain pH9_R2IIA[15] and compared the assemblies to publicly available genomes. Details on the genome analysis and results are given in Supplementary Note 1.

The antibiotic resistance genes (ARG) detected in the sequenced genomes and the inferred antibiotic resistances are summarized in Supplementary Fig. 9. These detected ARGs conformed for the most part with the results from the antimicrobial susceptibility tests; however, there were also some discrepancies. For example, both *Bacillus* strains had genes for the transcription-repair coupling factor *mfd* and the

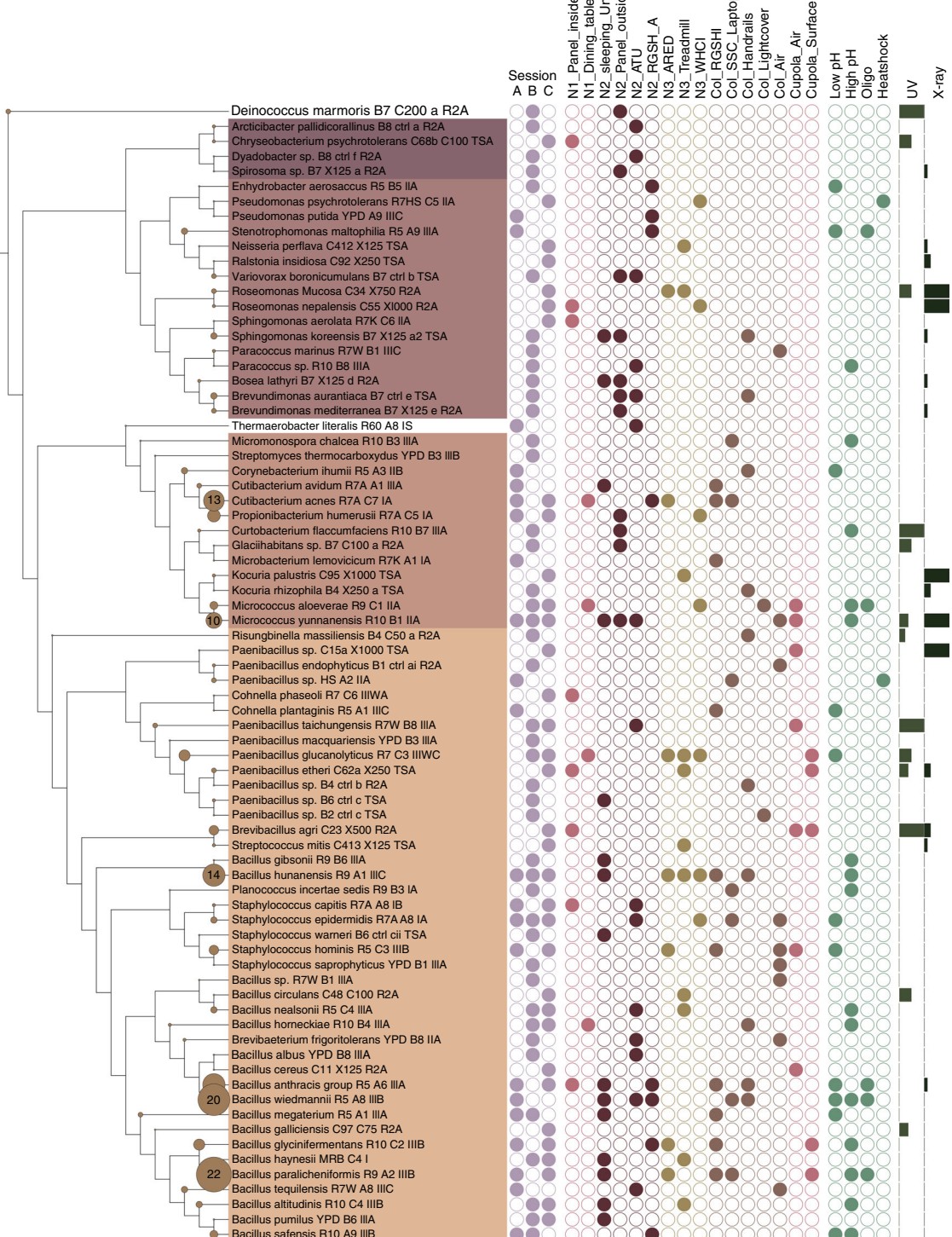

**Fig. 5** Tree of all isolates retrieved from sessions A–C. Circles on branches reflect the number of isolates obtained from the respective species. First dataset reflects the session number (A–C, blue), the locations where the isolates were retrieved from (Node (N) 1, 2, 3, Columbus (Col) and Cupola in different shades of brown), the enrichment conditions (low, high pH, oligotrophic conditions and heat-shock resistant; green) and the radiation dose (UVC, X-Ray; bars on the right; gray) which were used to pre-treat the samples

efflux transporter *blt* which should provide resistance against multiple fluorquinolones, but *Bacillus pumilus* strain pH7_R2F_2_A was only resistant against moxifloxacin and not against ciprofloxacin or levofloxain, while *Bacillus safensis* strain pH9_R2_5_I_C was sensitive against all tested fluorquinolones. *C. metallidurans* was unharmed by the lincosamide clindamycin and the oxazolidinone linezolid and grew at

the maximal tested concentrations of these antibiotics, but these resistances could not be inferred from the ARGs. Moreover, *C. metallidurans* was sensitive to all fluoroquinolones and β-lactam antibiotics besides Penicillin G in spite of possessing several efflux transporter genes from which a resistance against fluoroquinolones can be inferred and also the β-lactamase *ampC*, which is a specialized cephalosporinase

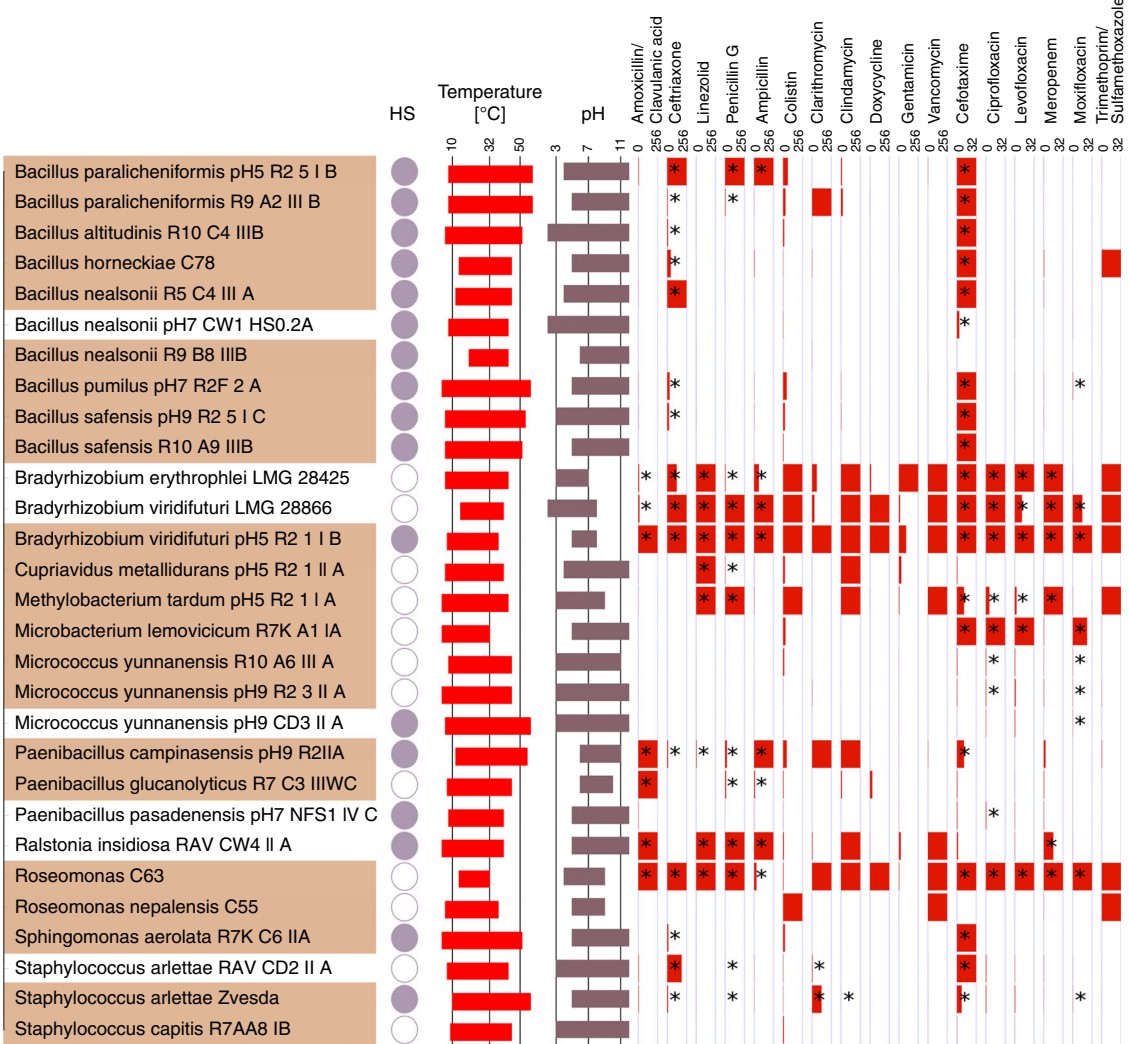

**Fig. 6** Resistances of selected microorganisms from ISS. ISS isolates (highlighted in brownish color) were compared with closely related isolates from ground controls (clean rooms, culture collections; white). Datasets reflect resistances to heat shock (filled circle: heat-shock resistance), temperature range of growth (red), pH range of growth (gray) and resistance toward antibiotics (dark red). Antibiotics were applied in maximal concentrations of 256 and 32 μg/ml, as indicated. Bars reflect the minimal inhibitory concentrations as determined experimentally. A full red bar indicates resistance against the maximal concentration tested. Asterisks indicate when an isolate was deemed resistant against a certain antibiotic according to the EUCAST guidelines for PK/PD (non-species related) and *Staphylococcus* spp. breakpoints. Partial information on some of the isolates is already available[15]

and infers resistance against the tested cefotaxime and ceftriaxone. However, it is known that these two cephalosporins, while being sensitive to *ampC*, are only weak inducers for actual *ampC* expression[27].

**Humans and the clean room as a contamination source.** The microbiome of an ISS cargo-harboring clean room, which we analyzed for comparative reasons, was found to be different from the ISS microbiome. The microbial diversity detected in clean rooms was significantly lower than observed in ISS samples (ANOVA; $p = 0.012$; Shannon diversity index) and clustered separately in multivariate analyses (Supplementary Fig. 10). The clean room microbiome was specifically characterized by a predominant abundance of α-Proteobacteria, such as *Novosphingobium*, *Sphingomonas*, and *Methylobacterium* whereas most ISS samples were dominated by Firmicutes and Actinobacteria. This is in accordance with previous findings[14]. Based on this observation, we argue that the items delivered from terrestrial clean rooms to the ISS are most likely not a (dominant) microbiome source.

However, a more detailed picture was obtained, when we looked at the cultivable diversity retrieved from ISS and clean room samples (Supplementary Fig. 11), where we found an overlap of several bacterial species, including *Bacillus cereus, B. aerophilus, B. subtilis, B. nealsonii, Micrococcus aloeverae, M. yunnanensis, Kocuria palustris,* and *Ralstonia insidiosa.* Human-associated microorganisms, such as e.g., *Micrococcus* species, were more likely introduced by humans into both environments than transported via cargo from clean rooms to the ISS. Nevertheless, this comparison indicates a potential transfer of clean room-associated microorganisms (such as *Bacillus* species) onto the ISS where they established themselves as a part of the ISS microbial community. Based on the total cultivable diversity they comprise only a minor fraction of the ISS microbial community.

**ISS possibly harbors previously undescribed microbial species.** The application of 21 different cultivation approaches resulted in a high diversity of microbial isolates from the ISS (see Supplementary Fig. 11), and 22 of the bacterial genera obtained during this study have not been isolated from ISS samples before,

although most of them have been detected by molecular methods[13,14,20] (*Arcticibacter, Bosea, Brevundimonas, Chryseobacterium, Cohnella, Curtobacterium, Cutibacterium, Deinococcus, Dyadobacter, Enhydrobacter, Glaciihabitans, Micromonospora, Neisseria, Paracoccus, Planococcus, Propionibacterium, Risungbinella, Roseomonas, Spirosoma, Stenotrophomonas, Thermaerobacter,* and *Variovorax*). Of our fungal isolates, only *Aspergillus unguis* was not found on the ISS before.

In addition, seven of the ISS isolates obtained in this study might even qualify to comprise novel, hitherto undescribed bacterial species as their 16S rRNA gene sequence similarity to their respective closest described neighbor was below 98% (see Supplementary Table 3).

**Microorganisms interact with ISS surfaces**. ISS isolates *Cupriavidus metallidurans* strain pH5_R2_1_II_A, *Bacillus paralicheniformis* strain R2A_5R_0.5[15], and *Cutibacterium avidum* strain R7A_A1_IIIA were aerobically and anaerobically incubated together with untreated aluminum alloy platelets, anodized aluminum alloy platelets, and pieces of NOMEX® fabric, to investigate if these isolates interact with these ISS surface materials. After incubation (6 weeks; 32 °C; liquid R2A medium), the co-incubated materials were analyzed via scanning electron microscopy (Fig. 7). The NOMEX® fabric itself remained intact over time (Fig. 7, I–L, negative control), but served as an excellent attachment surface for *B. paralicheniformis* biofilms and single cells of *C. metallidurans*. The ability of *C. metallidurans* to attach to surfaces was likely linked to its genetically encoded pili formation capacity (see above).

*B. paralicheniformis* did neither adhere to the untreated nor to the anodized aluminum alloy, as shown in Supplementary Figs. 12–14. However, the untreated aluminum alloy which was co-incubated with *B. paralicheniformis* showed sporadical signs of corrosion compared with the untreated negative control incubated in sterile medium (Supplementary Fig. 12). All anodized aluminum surfaces showed attached debris regardless if the incubation in the sterile medium was performed under oxic or anoxic conditions (Supplementary Figs. 13 and 14). Single cells of *C. metallidurans* also attached to the surfaces of untreated and anodized aluminum alloys and their co-incubated aluminum alloys had a unique background surface pattern, which was distinct from their respective negative controls (Supplementary Figs. 12 and 13). Under anoxic conditions, *C. avidum* formed a biofilm attached to the surface of both, untreated and anodized, aluminum alloys (Supplementary Fig. 14).

**Metagenomics reveals unique microbial composition and functions**. We compared our shotgun metagenomics data with datasets available, i.e., the Home Microbiome Project[28] and the Indoor Microbiology Project (ref. [5]; both publicly available through MG-Rast). The Home Microbiome Project contains metagenomics datasets from indoor surfaces (e.g., floors, door knobs, and light switches) as well as from human body sites (e.g., skin, nose, and foot). The Indoor Microbiology dataset contains metagenomics data from controlled (e.g., clean room) and uncontrolled (e.g., public, private houses) indoor surfaces.

The derived taxonomic diversity from the metagenomics dataset from our study showed the lowest diversity amongst all analyzed datasets (Shannon Index, $p = 0.00034$, ANOVA) and the microbial composition grouped between human- and indoor-samples. The grouping (ISS, indoor, and human) was found to be a significant factor ($p = 0.001$, RDA+). In a clustered barchart analysis (100 most abundant genera), the ISS samples grouped together with hand and bathroom door knob microbiomes (Fig. 8). LEfSe identified *Propionibacterium, Streptococcus,*

*Clostridium, Ralstonia, Bacteroides, Veillonella, Haemophilus, Enterococcus, Brevibacterium,* and *Rubrobacter* to be indicative for the ISS microbiome, when compared with human and indoor samples from the Home Microbiome Project (Supplementary Fig. 15).

Comparisons on functional level confirmed our observations, as resistance-associated gene signatures were not significantly increased compared with terrestrial indoor environments; in contrast, multidrug-resistance efflux pumps were found to be significantly reduced compared with terrestrial indoor environments and human samples (Fig. 9). However, the ISS microbiome showed increased signatures for ABC-type iron transport systems, cadmium resistance, and chromium compounds, confirming that ISS microbes are specifically adapted to surface/metal interaction.

Overall, genes involved in stress-response were even found to be significantly lowered in relative abundance compared with indoor environments on ground. The ISS microbiome, however, showed significantly increased levels of genes involved in dormancy and sporulation, and adhesion (compared with terrestrial indoor environments; Fig. 9). Notably, the shotgun metagenomics analyses revealed a higher alpha-diversity of functions aboard the ISS (Fig. 9b).

## Discussion

In this study, we aimed to exploit the microbial information obtained from three sampling events aboard the International Space Station with respect to: (i) microbial sources, diversity and distribution within the ISS, (ii) functional capacity of microbiome and microbial isolates, (iii) extremotolerance and antibiotics resistance (compared with ground controls), and (iv) microbial behavior toward ISS-relevant materials (biofilm formation, potential degradation, or corrosion).

Based on our observations and previous reports[12], we confirm a mostly human-associated microbiome aboard the ISS[29]. Other proposed sources are cargo delivery, food (such as seasoning, dried fruits, nuts and herbs, or even probiotics[30], as indicated by the presence of e.g., *Bacillus* and *Lactococcus* signatures), and potentially the personal belongings brought to the ISS. It shall be noted, that cargo deliveries are cleaned or even disinfected before upload[31,32], but an international standard for these procedures does not exist and thus the cleanliness of the cargo delivery might vary.

As a consequence, the ISS microbiome was characterized by a predominance of human (skin)- associated *Staphylococcus, Corynebacterium* and *Streptococcus* signatures[33]. In general, these microorganisms are typical indicators for confined indoor environments (clean rooms, space stations, hospital areas such as intensive care unit, operating rooms[12]; see also[14]).

Accordingly, all top 20 genera described in the hospital study[33] were also detected in the entire ISS microbiome (mostly also under top 20). *Enhydrobacter* (Pseudomonadales, a typical environmental species[34]) was the only hospital top 20 genus which was not detected by the molecular approach in this study, but an *Enhydrobacter aerosaccus* isolate was obtained from the Columbus RGSH. Within the top 20 list of the ISS microbial signatures, *Haemophilus, Aerococcus, Stenotrophomonas, Gemella, Bacteroides, Actinomyces, Veillonella, Granulicatella, Blautia, Propionibacterium,* and *Enterobacter* could not be found in the top 20 hospital list, indicating that those were more abundant in the ISS or even specific for this location.

All of these genera are typical human-associated microorganisms and thrive in the oral/respiratory tract (e.g., *Haemophilus*), on human skin (e.g., *Propionibacterium*) or the human gut (e.g., *Blautia*). Some of these microorganisms have opportunistic pathogenic potential, as also pointed out for *Enterobacter* species isolated from the ISS WHC[35]. We obtained in total eleven ISS

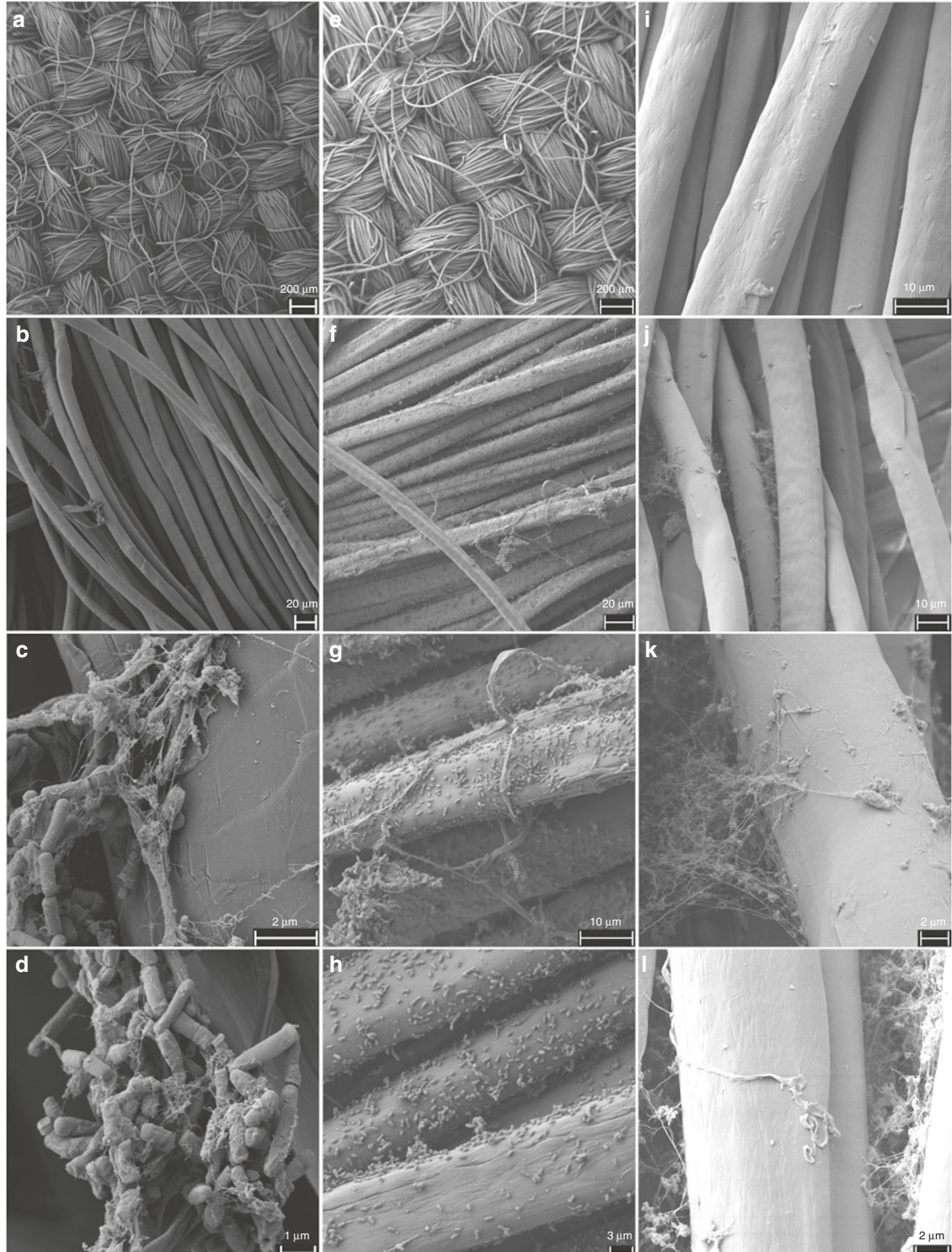

**Fig. 7** Scanning electron micrographs of NOMEX® fabric. The fabric was co-incubated for 6 weeks with bacteria isolated from the ISS: **a–d** Co-incubation with *Bacillus paralicheniformis*; **e–h** Co-incubation with *Cupriavidus metallidurans*; **i–l** Negative control of NOMEX® fabric kept in sterile medium for 6 weeks. Precipitates found in the negative control were found to be remnants of the medium/particles and did not contain microbial cells

isolates belonging to biosafety risk group S2 during this study, including *Pseudomonas putida* isolated from the RGSH in Node2 and isolates of the *Bacillus cereus/anthracis/thuringensis* clade retrieved from the RGSHs in Node2 and Columbus, from the hand grips in Columbus, and from the sleeping unit in Node2. However, most of these are typical human-associated bacteria which have only opportunistic pathogenic potential. Especially in

the light of a weakened human immune system in space conditions, the presence and abundance of such opportunistic pathogens has of course to be carefully monitored, but as these do thrive in and on the human body and are shed into the environment by the crew itself, such opportunistic pathogens will always exist in built environments and their presence *per se* is not alarming[12].

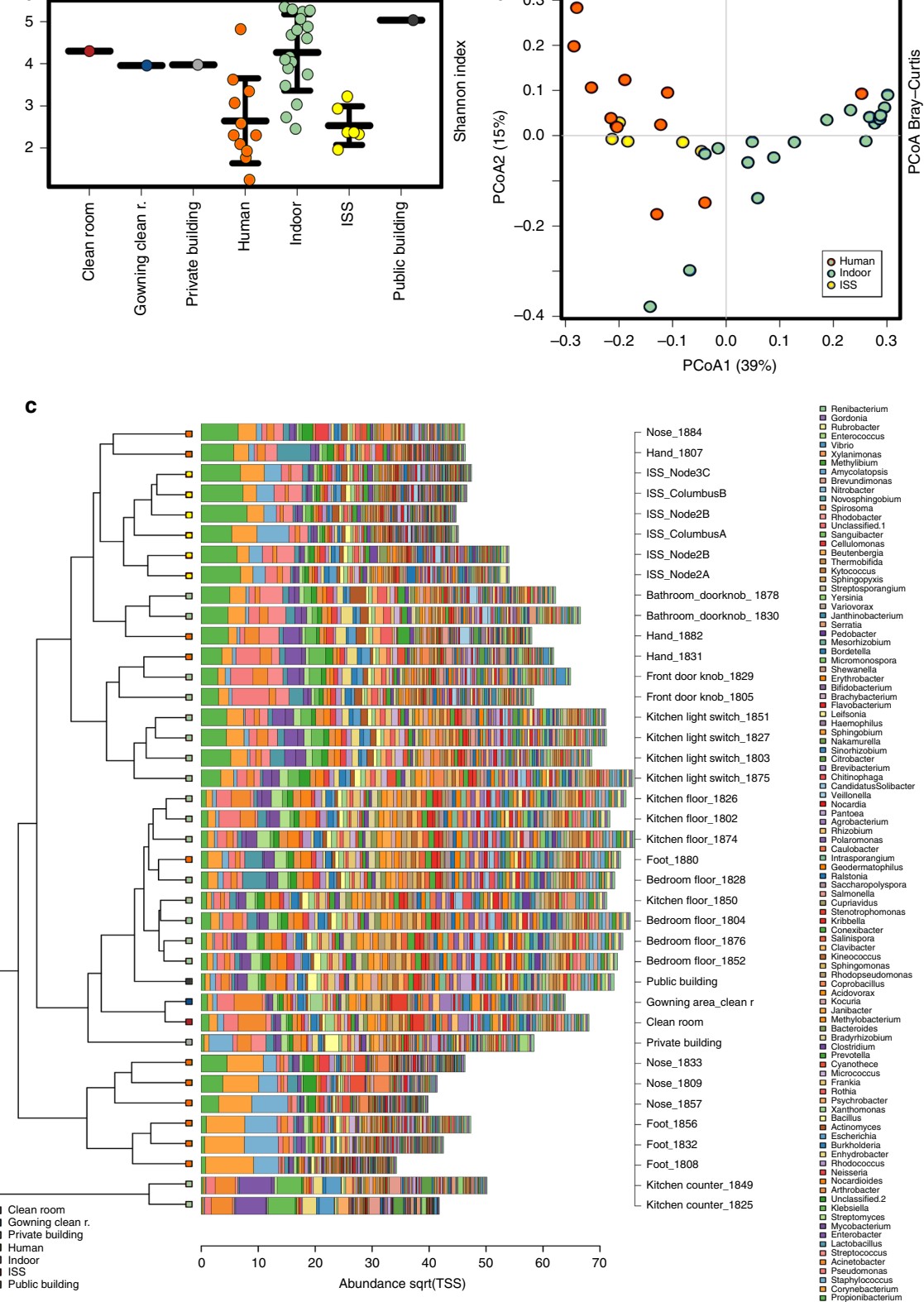

**Fig. 8** Shotgun metagenome-based comparison of ISS and terrestrial samples. Metagenomes- derived taxonomic information from terrestrial indoor environments (private homes, public buildings, clean rooms), human surface samples (nose, hand, foot) and samples from the ISS (public metagenomics information taken from refs. 5,28). **a** Alpha diversity (Shannon index); error bars reflect standard deviation. **b** PCoA plot, **c** Clustered barchart analysis

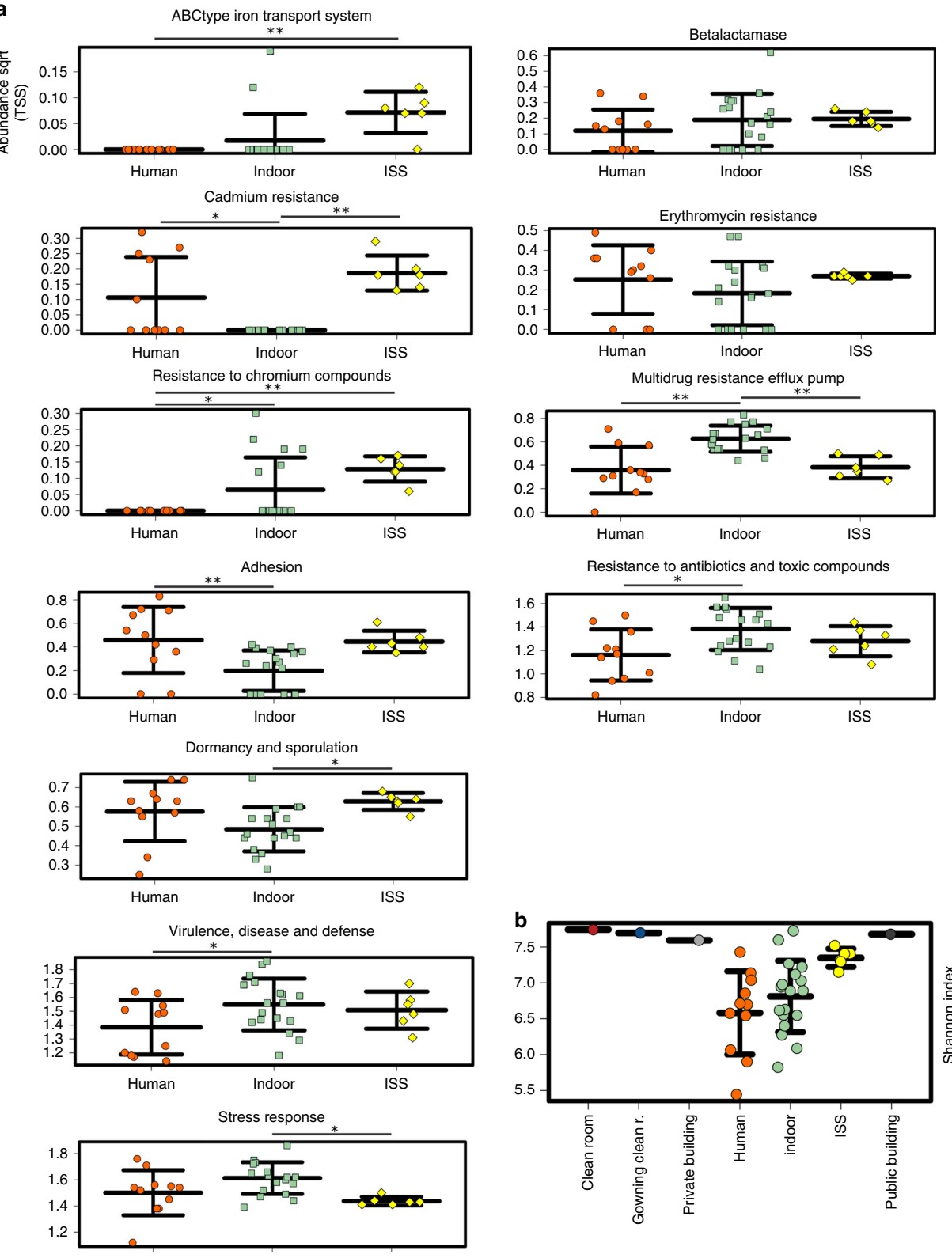

**Fig. 9** Functional characteristics. **a** Relative abundance of gene signatures of respective categories in the metagenomics datasets of the ISS (this study; yellow), terrestrial indoor (green) and human surfaces (orange). Statistical significance (*: $p < 0.05$; **: $p < 0.01$) was inferred using ANOVA, followed by a Tukey–Kramer post hoc test and Benjamini–Hochberg FDR correction. **b** Shannon diversity index of functions in different data sets. All stripcharts: error bars visualize standard deviation

The ISS microbiome was not found to be stable in composition and diversity, although a vast core microbiome existed over time and independent from location. All of the core microbiome genera have also been found in ISS dust samples from 2004 and 2008, as well as other ISS microbiome studies, indicating that this core microbiome is indeed established onboard the ISS, independent of the individual crew microbiome[15,20]. However, specific microbial patterns could be identified for various functional areas within the station, as e.g., the WHC and RGSH showed the highest number of unique RSVs. This is in agreement with the observations of Ruiz-Calderon et al. that, increasing with the level of human interaction, indoor surfaces reflect space use and, besides, show increased content of human-associated microbial signatures[36].

Differences in diversity and composition were detectable when two different time points (session A and B) were compared. It shall be mentioned, that no crew exchange took place during this period, but two cargo deliveries docked within this time frame, which could have influenced the microbiome composition. However, we detected an increase in specifically human (gut)-associated microorganisms (*Escherichia/Shigella*, *Lachnoclostridium* etc.) over the sampling period, of which the reason is unknown. In this context, future research is needed covering multiple time points over a larger time series (monitoring) to better understand the microbiome dynamics and adaptation, but also possible transmission from and to humans.

A different picture was obtained from the cultivable diversity of the ISS microbial community, with *Micrococcus yunnanensis*, *Bacillus hunanensis, B. megaterium* and *B. safensis,* and *Staphylococcus epidermidis* being cultivable from both sampling sessions, representing hardy (spore-forming) and human skin- associated microorganisms, whereas typical gut-associated microorganisms could not be retrieved by our enrichments, as our cultivation approaches were designed to target rather environmental, extremotolerant microbes. Thus, the cultivation- and molecular-based microbial community analysis is not fully comparable due to biases introduced by both methods. These include limitations in cultivation efforts, problems with DNA extraction and sequencing, and the detection of DNA also from nonviable organisms.

Archaeal signatures were detected in 14 of the 24 ISS samples (universal and specific approach). Most earlier studies on the ISS microbiome ignored the possible presence of Archaea, but some of the more recent studies also reported the detection of Archaea aboard the ISS but did not further discuss their existence[20,23]. We found, that archaeal signatures were nicely reflecting the frequency of human contact and the type of surface (see Fig. 5). The detected sequences of Thaumarchaeota, Woesearchaeota, and *Methanobrevibacter* (Euryarchaeota) have all been attributed to the human microbiome before[37–39].

Our results support the pangenome-based observation of Blaustein et al.[26], on genomic, but also on isolate and resistance-pattern level that ISS microorganisms are not necessarily more extremophilic or antibiotic resistant than their ground relatives. We propose that the ISS environment supports selection of the best-adapted microorganisms (e.g., spore-formers) toward the partially extreme physical and chemical environmental conditions (e.g., radiation, alkaline cleaning agents), but does not induce permanent changes in the physiological nor genomic capacities of microbes. Thus, we were not able to confirm the null hypothesis that strains obtained from the ISS are more extremotolerant/extremophilic than closely related strains from Earth regarding the upper and lower boundaries of their temperature and pH growth ranges. With the exception of *Bradyrhizobium viridifuturi* pH5_R2_1_I_B, all tested ISS isolates were able to grow at pH 9 or higher, which might be due to a selection pressure caused by alkaline cleaning reagents used onboard the ISS.

Moreover, the data presented here show that the molecular detection of antibiotic resistance genes, while being a good approximation of the resistance potential of an organism or microbial community, does on the one hand overestimate the antibiotic-resistance potential (as some resistance genes might not be expressed at all). On the other hand it does not necessarily cover all antibiotic resistances which a microorganism actually has. Thus we advise coupling traditional cultivation approaches with molecular investigations to retrieve a full picture of the situation.

Although we could not confirm an increased threat of ISS microbiome toward crew's health, we observed that surface interaction is critical for the microbial community aboard. A variety of ISS surface materials are composed of metals, including alloy EN AW 2219 (aluminum copper magnesium), which might cause stress in microorganisms during interaction with metal ions and settlement. In our co-incubation experiments, we could confirm that ISS microbial isolates can adhere and grow on metal and in particular textile surfaces (NOMEX® fabric), where local moisture (e.g., condensate) could support biofouling, biofilm formation, and material damage through acid production[6,40–42]. Surfaces onboard the ISS should of course not be exposed to nutrient rich liquid over such a long timeframe as it was done in this experiment, but our findings emphasize that local moisture has to be kept down to a minimum to prevent potentially harmful biofilm formation. This is additionally important with respect to fungal growth, as these might affect human health indirectly by causing allergic reactions and asthmatic responses[18,43–45].

Although we cannot fully exclude a threat of the ISS microbiome toward crew health (in particular in interaction with the weakened human immune system) our data do not indicate direct reason for concern. However, we raise special attention to the microbial-surface interaction problem in order to avoid biofouling and biofilm formation, which could directly impact material integrity and indirectly human health and therefore pose a potential risk to mission success.

## Methods

**Preflight preparations and sampling aboard the ISS**. Packaging, pre-processing and logistics of the sample material regarding upload and download from the ISS were managed by the Biotechnology Space Support Center (BIOTESC) of the Lucerne University of Applied Sciences and Arts (Switzerland). In-flight sampling aboard the ISS was performed during increment 51 and 52 (April to June 2017) under the ESA (European Space Agency) operation named EXTREMOPHILES. All samples were taken by US astronaut Jack D. Fischer. Sampling was performed in an area of ~1 m² for each location, either with dry wipes (session A and B) or premoistened wipes (session C; TX 3211 Alpha Wipe, ITW Texwipe, Kernersville, US, 23 × 23 cm; 20 ml autoclaved ultrapure water for chromatography, LiChrosolv, Merck Millipore). Three sampling sessions were performed, with session A on May 1st 2017, session B on July 12th 2017, and session C on June 29th 2017. With a span of 72 days between session A and B they were conceptualized for comparative sampling to assess microbiome fluctuation over time. An overview of all sampled areas and sessions is given in Table 1.

The sampling instructions for each session were as follows. (1) Put on sterile gloves (DNA-free nitrile gloves, ABF Diagnostics GmbH, Kranzberg, Germany). (2) Using gloved hand, remove wipe X from bag (metal closure bag, GML-alfaplast GmbH, Munich, Germany), wave wipe through the air (~20 s). Put wipe back into its bag and close properly. (3) Change glove. (4) Using gloved hand, take sample using wipe Y according to Table 1. Put wipe back into its bag and close properly. (5) Repeat steps 3 and 4 for every new sampling surface according to Table 1. (6) Store wipes at ambient (session A, B, dry wipes), or under cool conditions ("cold stowage", 2–10 °C, session C, moist wipes).

The sampling material was uploaded via a Cygnus transporter during Orbital ATK flight CRS-7 on April 18th, 2017 and downloaded via SpaceX cargo in June (session A + C) and September 2017 (session B).

**Clean room and cargo vehicle sampling**. In order to retrieve samples for comparative analyses, one ISS-relevant clean room and cargo-spacecraft was sampled, namely clean room S5C at the Centre Spatial Guyanais near Kourou in French Guiana, housing ATV5 "Georges Lemaître". Swab (FLOQSwabs™, Copan

**Table 1 Sampling locations and sampling sessions.** In total, 24 wipes were retrieved from 5 different modules, and 15 different locations within the ISS

| Wipe | Sampled surface | ISS module | Session |
|---|---|---|---|
| A-5, B-1 | Ambient air (field blank, FB) | Columbus | A, B |
| A-4, B-2 | Light covers | Columbus | A, B |
| A-2, B-3 | SSC laptop | Columbus | A, B |
| A-3, B-4 | Hand grips | Columbus | A, B |
| A-1, B-5 | Return Grid Sensor Housing (RGSH) | Columbus | A, B |
| A-6, B-6 | Sleeping unit | Node 2 | A, B |
| A-7, B-7 | Panels (outer surface, close to the Portable Fire Extinguisher (PFA) and Portable Breathing Apparatus (PBA)) | Node 2 | A, B |
| A-8, B-8 | Audio Terminal Unit (ATU) | Node 2 | A, B |
| A-9 | Return Grid Sensor Housing (RGSH) | Node 2 | A |
| C-1 | Ambient air (field blank, FB) | Cupola | C |
| C-2 | Surface facing a window | Cupola | C |
| C-3 | Advanced Resistive Exercise Device (ARED) | Node 3 | C |
| C-4 | Treadmill | Node 3 | C |
| C-5 | Waste and Hygiene Compartment (WHC): surfaces | Node 3 | C |
| C-6 | Cover of the PBA, inside | Node 1 | C |
| C-7 | Dining table | Node 1 | C |

diagnostics, USA) and wipe samples from ATV and its clean room were provided by Stefanie Raffestin (ESA) in 2014.

**Sample extraction.** The obtained sample material was either available as wipes or swabs (clean room). Wipes were submerged in 80 ml DNA-free 0.9% (w/v) NaCl solution (NaCl was heat treated to destroy DNA residues for 24 h, 250 °C), vortexed (10 s) and shaken manually (15 s), ultra-sonicated at 40 kHz for 2 min and vortexed (10 s). The sampling material was aseptically removed from the extraction solution before cultivation- and molecular analyses. Swabs were submerged in 15 ml of NaCl solution and processed identically.

**Cultivation.** Cultivation of microorganisms was performed on a number of solid and liquid media, as given in Supplementary Table 4. For microbial enrichment, we provided variable chemical and physical conditions with respect to: pH (4–10), temperature (4–65 °C), gas phase (aerobic, $N_2$:$H_2$:$CO_2$, $H_2$:$CO_2$, $N_2$:$CO_2$), nutrients and nutrient availability. R2A (pH 5-7), RAVAN and ROGOSA were supplemented with nystatin (50 µg/ml) to suppress growth of fungi; media targeting archaea were supplemented with 50 µg/ml streptomycin and 100 µg/ml ampicillin. Inoculation was done using 500 and 250 µl (duplicates) of the extraction solution. In addition, 500 µl aliquots of the wipe suspension were irradiated at the DLR in Cologne, Germany, to select for radiation resistant isolates. They were either irradiated by UV-C (254 nm) with an intensity of 50, 75, 100, and 200 J/m$^2$ or by X-rays with an intensity of 125 Gy, 250 Gy, 500 Gy, 750 Gy, or 1000 Gy. Radiation resistant microorganisms were cultivated on R2A and TSA agar. Pure cultures were obtained via repeated dilution series in liquid medium and/or purification streaks on solid media.

**ITS/16S rRNA gene sequencing and fungal/bacterial classification.** Partial 16S rRNA genes of the isolates were amplified using the primers 9bF (5′-GRGTTTGATCCTGGCTCAG-3′) and 1406uR (5′-ACGGGCGGTGTGTRCAA-3′), applying the following cycling conditions: Initial denaturation at 95 °C for 2 min, followed by 10 cycles of denaturing at 96 °C for 30 s, annealing at 60 °C for 30 s and elongation at 72 °C for 60 s, followed by another 22 cycles of denaturing at 94 °C for 30 s, annealing at 60 °C for 30 s and elongation at 72 °C for 60 s, and a final elongation step at 72 °C for 10 min[46]. The template was either a small fraction of a picked colony in a colony-PCR assay or 5–20 ng of DNA purified from culture via the peqGOLD Bacterial DNA Kit (Peqlab, Germany). The 16S rRNA gene amplicons were visualized on a 1.5% agarose gel, purified with the Min Elute PCR Purification Kit (Qiagen, Netherlands) or the Monarch PCR and DNA Cleanup Kit (New England Biolabs, US). After Sanger-sequencing (Eurofins, Germany) the obtained sequences were classified using the EzBioCloud platform at http://www.ezbiocloud.net/eztaxon[47].

The ITS region of fungal isolates was sequenced using the primers ITS1F (5′-CTTGGTCATTTAGAGGAAGTAA-3′) and ITS4 (5′-TCCTCCGCTTATTGATATGC-3′) and following cycling conditions: initial denaturation at 95 °C for 10 min, followed by 35 cycles of denaturing at 94 °C for

60 s, annealing at 51 °C for 60 s, elongation at 72 °C for 60 s, and a final elongation step at 72 °C for 8 min. The amplicons were Sanger-sequenced (Eurofins, Germany) and the obtained sequence was classified using the curated databases UNITE version 7.2[48] and BOLD version 4[49]. Fungal isolates of session A, B, and C were classified according to phenotypic characteristics

**Phylogenetic tree reconstruction.** For phylogenetic tree reconstruction, the forward and reverse sequences obtained from the isolates were merged to reach a minimum sequence length of 1000 bp. The phylogenetic tree was calculated with the Fast Tree program[50] and displayed with the Interactive Tree of Life online tool iTOL[51].

**DNA extraction of ISS wipe samples.** After aliquots were removed for cultivation assays, the rest of the wipe solutions were filled into Amicon Ultra-15 filter tubes (Sigma Aldrich) and were centrifuged at $4000 \times g$ for 10–30 min at 4 °C. The flow-through was discarded and the remaining liquid in the filters was pipetted into 1.5 ml Eppendorf tubes for DNA extraction with the modified XS- buffer method[52]. Briefly, samples were incubated with XS-buffer ($2 \times$, 20 ml stock solution: 1 M Tris/HCl (pH 7.4) (4 ml); 7 M ammonium acetate (4.56 ml); 250 m ethylene diamine tetraacetic acid (3.2 ml); 10% sodium dodecyl sulfate (w/v) (4 ml); potassium ethyl xanthogenate (0.4 g); PCR-grade water (4.99 ml)) for 2 h at 65 °C to open the cells. Phenol/chloroform extraction was used afterwards to isolate the DNA. DNA concentrations were determined using Qubit (High Sensitivity Kit, Life Technologies, US).

**Microbial profiling using next-generation sequencing methods.** To investigate the detectable molecular diversity, we used a "universal" and an Archaea-targeting approach. The 16S rRNA gene amplicons for the universal approach were amplified using Illumina-tagged primers F515 (5′- TCGTCGG-CAGCGTCAGA TGTGTATAAGAGACAGGTGCCAGCMGCCGCGGTAA-3′) and R806 (5′-GT CTCGTGGGCTCGGAGATGTGTATAAGAGACAGGGAC-TACHVGGGTWTC TAA-3′)[53]. Archaeal amplicons were obtained by a nested approach[54]: First, a ~550 bp-long 16S rRNA gene amplicon was generated with the primers Arch344F (5′-ACGGGGYGCAGCAGGCGCGA-3′) and Arch915R (5′-GTGCTCCCCCGCCA ATTCCT-3′)[55,56] and in a second PCR, the amplicons for Illumina sequencing were generated by the tagged primers Arch519F (5′-TCGTCGGCAGCGTCAGA TGTGTATAAGAGACAGCAGCMGCCGCGGTAA-3′) and Arch785R (5′-GTCT CGTGGGCTCGGAGATGTGTATAAGAGACAGGACTACHVGGGTATCTAAT CC-3′)[57], using the purified product of the first PCR as template. The cycling conditions for the universal approach were initial denaturation at 94 °C for 3 min, followed by 35 cycles of denaturing at 94 °C for 45 s, annealing at 60 °C for 60 s and elongation at 72 °C for 90 s, followed by a final elongation step at 72 °C for 10 min. For the first PCR of the nested archaeal approach, the cycling conditions were initial denaturation at 95 °C for 2 min, followed by 10 cycles of denaturing at 96 °C for 30 s, annealing at 60 °C for 30 s, and elongation at 72 °C for 60 s, followed by another 15 cycles of denaturing at 94 °C for 30 s, annealing at 60 °C for 30 s, and elongation at 72 °C for 60 s, and a final elongation step at 72 °C for 10 min. For the second amplification the cycling conditions were initial denaturation at 95 °C for 5 min, followed by 25 cycles of denaturing at 95 °C for 40 s, annealing at 63 °C for 120 s and elongation at 72 °C for 60 s, followed by a final elongation step at 72 °C for 10 min.

**Genome sequencing, genome reconstruction, and annotation.** We sequenced the genomic DNA of six isolates obtained from ISS samples described earlier[15]. DNA was isolated from overnight cultures using the peqGOLD bacterial DNA mini kit (Peqlab, Germany). Double stranded DNA was quantified via Qubit Fluorometer 2.0 (Invitrogen, USA) according to the manufacturer's instructions. Library preparation and sequencing was carried out at the Core Facility Molecular Biology at the Center for Medical Research at the Medical University Graz, Austria.

Genomic reads were quality checked with FastQC[58] and then filtered with Trimmomatic (removed all adapter sequences, SLIDINGWINDOW 4:20, MINLEN 50)[59]. Genomes were assembled with SPADES in careful mode[60] and afterward checked for completeness via CheckM[61]. The assemblies were annotated and compared with closely related reference strains via the microbial genome annotation & analysis platform MicroScope (http://www.genoscope.cns.fr/agc/microscope)[62–64].

**Resistance and physiological tests.** Experiments were performed with selected microbial isolates from this and our recent study on ISS microorganisms[15]. (i) **Heat- shock resistance test**: The heat-shock test was carried out according to ESA standards[65]. In brief, single colonies of 3–5-day old cultures were suspended in two test tubes containing 2.5 ml sterile PBS. One tube was incubated at room temperature (control), whereas the other was placed in a water bath and exposed for 15 min to 80 °C. Samples were immediately cooled down on ice for 5 min after incubation time. The temperature was monitored using a separate pilot tube containing 2.5 ml of PBS. Afterward, 0.5 ml of the heat-shocked suspension and 0.5 ml of the room temperature suspension were plated and incubated at 32 °C for 72 h. (ii) **Physiological tests**: For the assessment of the temperature range, cultures were plated on R2A pH7 agar and incubated overnight at 32 °C. Then the

incubation temperatures for the species still growing were stepwise decreased and increased until no further growth was observed. Limits of pH tolerance were assessed accordingly. (iii) **Antibiotics susceptibility tests**: Antimicrobial susceptibility testing for selected, clinically relevant antibiotics (Supplementary Table 5) was performed using Etest® reagent strips (Biomérieux, Germany) according to manufacturer's instruction and detailed in[15]. Since there were no species-specific breakpoints available, MICs were interpreted according to EUCAST guideline table "PK/PD (Non-species related) breakpoints"[66]. The used PK/PD breakpoints in µg/ml, as given in the EUCAST breakpoint tables v8.1, were: amoxicillin/clavulanic acid: S ≤2, R >8; ampicillin: S ≤2, R >8; cefotaxime: S ≤1, R >2; ceftriaxone: S ≤1, R >2; ciprofloxacin: S ≤0.25, R >0.5; levofloxacin: S ≤0.5, R >1; linezolid: S ≤2, R >4; meropenem: S ≤2, R >8; moxifloxacin: S ≤0.25, R >0.25; penicillin G: S ≤0.25, R >2. The defined *Staphylococcus* spp. breakpoints used for the tested *Staphylococcus* strains differ from the PK/PD breakpoints regarding the following antibiotics: ciprofloxacin: S ≤1, R >1; clarithromycin ≤1, R >2; clindamycin S ≤0.25, R >0.5; doxycycline S ≤1, R >2; gentamicin: S ≤1, R >1; levofloxacin: S ≤1, R >1; linezolid: S ≤4, R >4; penicillin G: S ≤0.125, R >0.125; trimethoprim/sulfamethoxazole: S ≤2, R >4; vancomycin: S ≤4, R >4. For colistin, clarithromycin, clindamycin, doxycycline, gentamicin, vancomycin, and trimethoprim/sulfamethoxazole no defined PK/PD breakpoints exist.

In brief, overnight cultures (2–3 day cultures for slower-growing bacteria) were suspended in 0.9% saline. One hundred microliters of this suspension was plated on standardized Müller–Hinton agar for antimicrobial susceptibility testing (Becton Dickinson, USA). Etest® reagent strips were placed on the plates followed by aerobic incubation for 24 h at 34 °C.

**Co-incubation experiments and electron microscopy**. To test if some of our isolates interact with, and possibly damage, materials used aboard the ISS, we incubated them together with relevant ISS material. Pieces of NOMEX® fabric were provided by the BIOTESC of the Lucerne University and plates of the aluminum copper magnesium alloy EN AW 2219 which is also used on the ISS, were provided by Thales Alenia Space (TAS), Italy. NOMEX® is a flexible, flameproof fabric used for most storage bags aboard the ISS. The NOMEX® fabric was cut into pieces of 20 mm × 30 mm and autoclaved before incubation. The aluminum alloy EN AW 2219 was cut into small plates of 20 mm × 30 mm × 3 mm by Josef Baumann in Falkenberg, Germany, and then evenly polished with a grit size of P240 and partly anodized by Heuberger Eloxal, Austria. The autoclaved metal platelets, non-anodized and anodized, and NOMEX® fabric pieces were then incubated together with bacteria isolated from the ISS: *Cupriavidus metallidurans* pH5_R2_1_II_A (aerobic), *Bacillus licheniformis* R2A_5R_0.5 (aerobic), and *Cutibacterium avidum* R7A_A1_IIIA (anaerobic). Incubations were done in triplicates over a period of 3 months in liquid R2A medium in Hungate tubes at pH7 and 32 °C. Every 2 weeks, 50% of the medium was exchanged to ensure survival and further growth of the bacteria. After incubation, metal plates and NOMEX® fabric pieces were investigated via scanning electron microscopy. Metal plates and NOMEX® fabric pieces from the co-incubation experiment with selected bacteria were aseptically removed from their respective Hungate tube, carefully rinsed with 1×PBS buffer and then fixated overnight in a 100 mM sodium cacodylate buffer containing 2.5% (v/v) glutaraldehyde at 4 °C. Scanning electron microscopy of the samples was performed at the Biocenter of the Ludwig-Maximilians-University Munich using a Zeiss Auriga cross beam unit (Zeiss, Oberkochen, Germany).

**Amplicon sequencing**. Library preparation and sequencing were carried out at the Core Facility Molecular Biology at the Center for Medical Research at the Medical University Graz, Austria. In brief, DNA concentrations were normalized using a SequalPrep™ normalization plate (Invitrogen), and each sample was indexed with a unique barcode sequence (eight cycles index PCR). After pooling of the indexed samples, a gel cut was carried out to purify the products of the index PCR. Sequencing was performed using the Illumina MiSeq device and MS-102-3003 MiSeq® Reagent Kit v3-600cycles (2 × 251 cycles).

**Sequence data processing and analysis**. Demultiplexed, paired reads were processed in R (version 3.2.2) using the R package DADA2 as described[67]. In brief, sequences were quality checked, filtered, and trimmed to a consistent length of ~270 bp (universal primer set) and ~140 bp (archaeal primer set). The trimming and filtering were performed on paired-end reads with a maximum of two expected errors per read (maxEE = 2). Passed sequences were dereplicated and subjected to the DADA2 algorithm to identify indel-mutations and substitutions. The DADA2 output table is not based on a clustering step and thus no operational taxonomic units (OTUs) were generated. Each row in the DADA2 output table corresponds to a non-chimeric inferred sample sequence (ribosomal sequence variants; RSVs)[67]. In addition, the merging step occurs after denoising, which increases accuracy. After merging paired end reads and chimera filtering, taxonomy was assigned with the RDP classifier and the SILVA v.123 trainset[68]. The visualization was carried out using the online software suite Calypso[69]. For bar plots data was normalized by total sum normalization (TSS) and for PCoA by TSS combined with square root transformation. For the Shannon index analysis, sequences were rarefied as indicated in the Figure captions. Tax4fun was performed based on the Silva-classified OTU table, as described[70].

**Microbial community network**. G-test for independence and edge weights were calculated on the RSV table using the make_otu_network.py script in QIIME 1.9.1[71]. The network table with calculated statistics was then imported into Cytoscape 3.7.1[72] and visualized as a bipartite network of sample (hexagons) and RSV nodes (circles) connected by edges. For clustering, a stochastic spring-embedded algorithm based on the calculated edge weights was used. Size, transparency and labels were correlated with RSV abundances, border line intensity refers to RSV persistence over multiple sampling sessions and edge transparency was correlated to calculated edge weights.

**Shotgun metagenomics**. Shotgun libraries for Illumina MiSeq sequencing were prepared with the NEBNext® Ultra II DNA Library Prep Kit for Illumina® in combination with the Index Primer Set 1 (NEB, Frankfurt, Germany) according to manufacturer's instructions and as described in[73]. Briefly, 500 ng of dsDNA were randomly fragmented by ultrasonication in a microTUBE on a M220 Focused-ultrasonicator™ (Covaris, USA) in a total volume of 130 µl 1×TE for 80s with 200 cycles per burst (140 peak incident power, 10% duty factor). After shearing, 200 ng of sheared DNA were used for the end repair and adapter ligation reactions in the NEBNext® Ultra II DNA Library Prep Kit for Illumina® according to manufacturer's instructions. Size selection and purification were performed according to the instructions for 300–400-bp insert size. Subsequent PCR amplification was performed with 4 cycles and libraries were eluted after successful amplification and purification in 33 µl 1×TE buffer pH 8.0. For quality control libraries were analyzed with a DNA High Sensitivity Kit on a 2100 Bioanalyzer system (Agilent Technologies, USA) and again quantified on a Quantus™ Fluorometer (Promega, Germany). An equimolar pool was sequenced on an Illumina MiSeq desktop sequencer (Illumina, CA, USA). Libraries were diluted to 8 pM and run with 5% PhiX and v3 600 cycles chemistry according to manufacturer's instructions. Raw fastq data files were uploaded to the metagenomics analysis server (MG-RAST)[74] and processed with default parameters. Annotations of taxonomy (RefSeq) and functions (subsystems) were then imported to QIIME 2 (2018.11)[75] or Calypso[69] to calculate core features, alpha and beta diversity metrics, statistics, and additional visualizations of the data sets. Additional statistical analyses were performed using STAMP[76].

**Controls**. Cultivation, extraction, PCR, and sequencing controls were processed and analyzed in parallel to biological samples. An unused wipe not taken out of its bag on the ISS was extracted for every sampling session, cut into pieces, placed on the different media and DNA was also extracted from the solutions obtained with the negative controls. All cultivation controls were negative (no growth of colonies). Wipe solutions of the negative controls used for DNA extraction, PCR, and sequencing revealed a low number of ribosomal sequence variants. These RSVs were removed from all other datasets, if present in the samples. In parallel, we used decontam[77] to purify the RSV table from possible "kitome" signatures[78]. Information on the decontam analysis and according results is provided in Supplementary Note 2, Supplementary Figs. 16–20, and Supplementary Data 5.

**Study limitations**. Spaceflight experiments especially suffer from limitations given by circumstances that cannot be influenced by the scientists. This affected e.g., the number of samples and replicates to be taken (limited mass of payload), the sampling procedure (compatible to microgravity conditions and safety requirements), selection of sampling time points (according to assigned crew time and the overall mission planning), and the delivery duration of the samples to the laboratory. Being aware of these circumstances, experiments were planned accordingly (>5 years implementation phase), and numerous controls were run to ensure the integrity of the information retrieved.

**Reporting summary**. Further information on research design is available in the Nature Research Reporting Summary linked to this article.

## Data availability
The assembled genomes are publicly available on the MicroScope platform[64] (*Bacillus pumilus* pH7_R2F_2_A; *B. safensis* pH9_R2_5_I_C; *Bradyrhizobium viridifuturi* pH5_R2_1_I_B; *Cupriavidus metallidurans* pH5_R2_1_II_A; *Methylobacterium tardum* pH5_R2_1_I_A; *Paenibacillus campinasensis* pH9_R2IIA) (http://www.genoscope.cns.fr/agc/microscope/mage/; type strain abbreviation in "genome browser" search window, no log-in required; data can be downloaded using the "Search/Export" tool). Raw 16S rRNA gene sequences of the molecular approach are available in the European nucleotide archive (https://www.ebi.ac.uk/ena), study ID: PRJEB30994. Metagenomic data are publicly available via MG-RAST (https://www.mg-rast.org/; MG-RAST ID: mgm4821480.3; no log-in required to access the data). The partial 16S rRNA gene sequences of unique bacterial isolates obtained during the EXTREMOPHILES flight project and from the clean room in Kourou are available at GenBank via accession numbers LR215073 to LR215191.

In addition, source data for analyses and figures are provided in Supplementary Data 1–5 as follows: Supplementary Data 1 (RSV table; Figs. 1–3, Supplementary Figs. 1–3, 5, 10), Supplementary Data 2 (metagenomic-based taxonomic profile; Fig. 4, Supplementary Fig. 7), Supplementary Data 3 (metagenomic-based functional profile; Fig. 9, Supplementary Fig. 8), Supplementary Data 4 (Information on isolates, Fig. 5),

and Supplementary Data 5 (Decontam dataset). The source data for Fig. 7 (original scanning electron micrographs) are provided as a Source Data file.

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

## Acknowledgements

We are thankful for financial support by the FFG (Austrian Research Promotion Agency, Project No. 847977) and thank the European Space Agency (ESA) for financing and realization of this space project as part of the ELIPS program (ILSRA-2009-1053 ARBEX). We are very grateful for the scientific and technical support by Lobke Zuijderduijn (ESA), Stefanie Raffestin (sampling in Kourou), the BIOTESC team (Lucerne University of Applied Sciences), astronaut Jack Fischer and all other involved members of space agencies and attached institutes. We thank R. van Houdt for providing a *Staphylococcus arlettae* isolate. The authors acknowledge the support of the ZMF Galaxy Team: Core Facility Computational Bioanalytics, Medical University of Graz, funded by the Austrian Federal Ministry of Education, Science and Research, Hochschulraum-Strukturmittel 2016 grant as part of BioTechMed Graz. We thank Thomas Rattei (Computational Systems Biology (CUBE), University of Vienna) for scientific discussion and computational support. PhD student Maximilian Mora was supported by the local PhD program MolMed and some of the results presented here are also published in M. Mora´s PhD thesis on the website of the Medical University Graz. C. S.C. acknowledges funding support from the Science and Technology Facilities Grant (No. ST/R000875/1).

## Author contributions

M.M., L.W., and I.K. conceptualized and performed experiments, wrote, and edited the paper. A.M. contributed to metagenomics analysis. P.R., P.S., C.S.C., and A.Z. contributed experiments with respect to physiological capabilities of ISS isolates. R.D. supported the realization of EXTREMOPHILES and implementation of the space experiment. T.A. and A.A. provided taxonomic classification of ISS fungal isolates. A.K. performed scanning electron microscopy. R.K. supported antibiotics resistance testing of ISS isolates. C.M.E. initiated and conceptualized the study, supervised the activities, analyzed data, and wrote the paper. All authors critically read the paper.

## Additional information

**Competing interests:** The authors declare no competing interests.

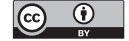

