## [Peer Review File · Nature Communications]

Reviewers' comments:

Reviewer #1 (Remarks to the Author):

The authors provide a detailed investigation of indoor environment samples from ISS, and explore variability over time, interaction of microbes with the ISS materials, and other aspect of indoor microbiome analysis. My expertise is in microbiome composition analysis from marker gene and metagenome data, and I have a number of concerns about those analyses that are essential to have addressed.

It's unclear why the authors sometimes using PCoA (line 302) and sometimes use NMDS (line 458). Can the authors clarify, or choose one method over the other? These are closely related methods, so switching between them doesn't immediately make sense.

Line 307: QIIME 1.9.1 has been replaced by QIIME 2 over a year ago. As far as I know, the functionality that the authors are using here doesn't exist in QIIME 2, but there surely must be more reliable methods for computing correlation networks now that are less prone to false positive associations. The authors should use an alternative approach for this. I believe CoNet is available in CytoScape, which the authors are already using for visualization. This paper provides a good discussion of more recent methods for network analysis:

<https://www.nature.com/articles/ismej2015235>

Line 366: This is too simplistic of an approach for handling false positive detections as RSVs present in negative controls may also truly be present in the study samples. For example, if there was human contamination of the negative controls, human skin microbes would be filtered from the analysis, which would preclude the observation that many of the ISS surfaces were dominated by human-associated microbes. Was this filter applied before other analyses, or only for this analysis? If only for this analysis, the authors should report the values with and without the filter applied. If for all analyses, the authors should use a better approach for integrating negative controls, if they seem important to integrate. Perhaps this method would be useful:

https://benjjneb.github.io/decontam/vignettes/decontam_intro.html

Line 397: The rarefaction process is not clearly explained here. Usually this is performed to achieve an even number of sequences per sample, not RSVs per sample. The description seems to be wrong as in the previous sentence the authors say they detected differences in the number of RSVs, but then say that all samples were subsampled to have the same number of RSVs. I think this is just an error in the description provided here. If the samples were rarefied to 649 sequences per sample, that is a very low number that should be clearly justified. With the technologies used here, we would typically see somewhere on the order of tens of thousands of sequences per sample.

Some important questions remain about the change in microbiome composition over time in this data. Is variability over time on ISS significantly greater than variability over time on the most similar ground-based indoor environment surfaces, or is this degree of variation to be expected? Was the degree of variation over time greater than or less than the variation across sites from the same sampling event?

Line 479: "besides potentially food for Lactococcus)." it's not clear what the authors mean here.

Line 727-737: "The ISS is a source of novel microbial species" This is overstated - it is a big jump to claim that these are newly identified species. Please provide the sequences for these eleven organisms, and the results from BLASTing them against the NCBI databases.

Minor comments:

Line 64: "loose microbial diversity" should be "lose microbial diversity"

Line 297: definition of an RSV does not imply a "separate taxonomic classification". For example, organisms of the same species or strain can have some variation in their rRNA. This clause of the sentence should be removed.

Reviewer #2 (Remarks to the Author):

Mora et al. report on the International Space Station experiment EXTREMOPHILES. This work advances understanding of the ISS microbiome structure and function by, in a novel way, combining molecular (amplicon sequencing, metagenomic, whole genome sequence) and cultivation (enrichment, physiology, resistance screen, microscopy) analyses. The manuscript is timely, generally well written, puts forth important findings, and will be of interest to a broad audience (e.g., microbiology, space travel, public health, engineering). However, there are several issues that must be addressed, mostly to improve the introduction and discussion, prior to publication.

Comments:

- The concept of the core ISS microbiome is intriguing, especially across the 5 ISS modules sampled. However, given the relatively short time frame of the study (May-July of 2017), the claim that the core is present over time becomes speculative. The discussion of this finding in context of previous work needs some improvement. Have all members of the core microbial community (all >50 genera) been detected in previous ISS expeditions with microbiome reports as well (e.g., Be et al. doi:10.1186/s40168-017-0292-4; Lang et al. doi:10.7717/peerj.4029)? This is important to support the claim of a core over time. Although you compare the "top 20" to the hospital microbiome study, it should also be compared to prior ISS expeditions. If the same core is not present, please explain why (perhaps human occupancy-related or possibly sequencing/primer bias) or change your presentation to describe a core microbiome within the time frame of the study. Would you expect "core" ISS microbial populations to change when a new crew arrives (could be an avenue for future research to present in the discussion)? Regarding all the above, please expand discussion following L871. Modify L 37-38, if needed, as well.
- The introduction should be restructured to link some of the brief statements/paragraphs together. For example, human health, dependency on microbiome, and both being compromised in space seem go together. Then get into ISS microbiome concerns – including pathogens, resistance, materials damage. That can be followed with the ISS description as a sealed off vessel and little knowledge on microbiome dynamics in an enclosed built environment, let alone with spaceflight conditions. Piecing together the ideas will improve the flow of the story.
- L76-77: What were the problems? What happened? Be more specific
- L96 and L101: Please don't start paragraphs with "publications focused on". Instead just get to the key point and cite the source.
- L109: Is "cosmonauts" necessary? Are they the same thing? Maybe change to "astronauts/cosmonauts"
- L112-116: Please restructure to form a stronger conclusion to the paragraph and overall background section. Try... "This may be attributable to environmental contamination limits (air, surfaces) having rarely been exceeded, at which times effective countermeasures had been implemented (24). Moreover, a genomics-based meta-analysis demonstrated that although pangenomes of Bacillus and Staphylococcus isolated from the ISS differed from Earth-based counterparts, these differences did not appear to be health-threatening (25). A comprehensive understanding of potential microbial adaptations to the ISS based on complementary genomic and cultivation approaches is needed to better understand potential human health implications (e.g.,

antibiotic resistance dissemination)."

- L118-123: Re-write section as the objectives of the study.
- L305: Re-name: "Microbial Community Network" or "Microbial Interaction Network"
- L343: Explicitly state a lack of sample replication (if that is true). It's not just the number of samples taken, but also the number of replicates. Replicates are critical for microbiome studies and a lack may also explain drastic differences across location and time point.
- L355-362: This information seems redundant with the methods. Remove.
- Fig. 2: Panel A should say "RSV" not "OTU". The caption should be more succinct and can begin with: "Temporal transitions in microbiome taxonomic diversity (same locations were sampled in Session A and Session B)."
- L392-408 and throughout the text: I think "dynamics" is a little too broad a term to express differences in only two points, especially considering limited replicates. Just be specific and say the microbiome changed between time A and time B. Please make corrections throughout the text to be more tentative on how "dynamic" the microbiome is, and just describe that it is subject to change. Again, the discussion could use some serious expansion on the role of microbial transfers in maintaining the microbiome. Also, how is microbial survival in the built environment (not necessarily the ISS) known to vary (at least for the dominant taxa that you identified)? Look to survival studies. This has implications for microbiome structure dependency on time.
- L452: Could be re-named to "Local factors shape microbiome composition". Also, open the section describing how local factors may actually vary to give premise for why locations have different microbiomes (e.g., human traffic, human activity, water, cleanliness and cleaning frequency, materials present)
- L479: This statement may be over-reaching as some environmental strains are clearly present too (e.g., *Bacillus*, *Pseudomonas*). Be specific and say that the majority of strains from the sequencing-based analysis appear human-derived.
- Fig 5: Are these all significant? Please indicate which correlations are actually significant in the chart
- L524-525: Where is this data presented? If nowhere, please add a supp figure.
- L534: Supp Fig 7 needs larger font. Also, consider presenting as a simple heatmap.
- L552: Fig. 7 needs a scale for node size
- L593, L597: need to add p-values
- L711: change "(relevant)" to "dominant" as they may still be relevant.
- L714: should be Fig. S9 not S6
- L751-2: change "was found" to "was likely linked to its" and remove "by"
- L760: did not adhere to
- L815-816: Split to a new paragraph. L815 will need a concluding sentence.
- L810-820: What statistical method was used to compare the functional attribute relative abundances – ANOVA, as described in Fig 12? If so, was an appropriate post-hoc p-value correction performed, such as Bonferroni or B-H False Disc Rate? Please implement a statistical test to remove potential false positives from the broad dataset in which abundance values are intrinsically dependent on all other values in the data set. Also, the metagenome function data needs to be made available in supplementary material, just like the RSV Taxa Table S1.
- L836: remove the parenthesis statement as this probably has to do with more human microbes from more frequent human occupancy
- L877: change "fluctuations" to "differences". Fluctuations implies that it changes over time, but it is really only two time points.
- L882: Paragraph needs a stronger conclusion. Lead into future need for research at multiple points over a larger time series to better understand microbiome evolution, transmission to humans, and health implications over time.
- L889: What about limitations and bias in sequencing efforts as well? Please indicate. Also, it is important to note that DNA detection does not mean viability.
- L899: "In our study we confirmed" should be changed to "Our results support" or "Our results are in line with" Also, this is a run-on sentence. Please modify.
- L919: How realistic was this simulation in relevance to bacteria degrading materials on the ISS? The inoculum was incubated with liquid media and a material for ~3 months (L274)? Seems

excessive. Are there any spaces aboard that are completely submerged, and if so, for how long (is biofouling really an issue or just biofilms)? Put your results in this context and just emphasize the necessity to keep moisture down to prevent biofilm growth and, in the events that biofilm forms, potential subsequent issues with material damage may arise.

- Supp Fig. S5: OTU should be changed to RSV
- Throughout the text, clarity in some of the figures would benefit from larger font size
- Fig 12: Perform Tukey post-hoc test on each ANOVA to indicate which of the three are same or different

Dear reviewers,

We highly appreciate your valuable comments! We addressed all comments and concerns in the rebuttal letter below and in the new version of the manuscript.

Thank you very much,

Christine Moissl-Eichinger and co-authors.

Reviewers' comments:

Reviewer #1 (Remarks to the Author):

The authors provide a detailed investigation of indoor environment samples from ISS, and explore variability over time, interaction of microbes with the ISS materials, and other aspect of indoor microbiome analysis. My expertise is in microbiome composition analysis from marker gene and metagenome data, and I have a number of concerns about those analyses that are essential to have addressed.

We are very thankful for the detailed comments and questions on our methodology. We addressed all your questions below.

☹ It's unclear why the authors sometimes using PCoA (line 302) and sometimes use NMDS (line 458). Can the authors clarify, or choose one method over the other? These are closely related methods, so switching between them doesn't immediately make sense.

It is correct that both methods are closely related, and there was no specific reason for choosing either PCoA or NMDS. Basically, the type of display for the NMDS in Fig.4 was allowing us to show better that the groups are not clearly separating from each other. For your reference, we add the PCoA here, but would prefer to keep the NMDS.

PCoA plot of the same data as used for Fig. 4 (NMDS).

☞ Line 307: QIIME 1.9.1 has been replaced by QIIME 2 over a year ago. As far as I know, the functionality that the authors are using here doesn't exist in QIIME 2, but there surely must be more reliable methods for computing correlation networks now that are less prone to false positive associations. The authors should use an alternative approach for this. I believe CoNet is available in CytoScape, which the authors are already using for visualization. This paper provides a good discussion of more recent methods for network analysis:

<https://www.nature.com/articles/ismej2015235>

First functionalities to compute co-occurrence networks in QIIME 2 (q2-SCNIC) were just integrated recently. Therefore, we focused on a reliable method, compatible with prior conducted network analysis through QIIME 1.9.1. Our attempt was to show shared RSVs between different sampled location of the ISS, with some statistical support. Running an analysis in CoNet would give us another perspective on the data, especially on positive and negative interactions between different features. However, this was not the main goal of this analysis. In contrast we were interested in comparability to network analysis we published before (e.g. Moissl-Eichinger et al. 2015, Mahnert et al. 2015, Mahnert et al. 2018, Mahnert et al. 2019) and a summarized view on relative abundances of shared taxa. However, we are very thankful for your comment and will consider recent methods for network analysis in future studies.

☞ Line 366: This is too simplistic of an approach for handling false positive detections as RSVs present in negative controls may also truly be present in the study samples. For example, if there was human contamination of the negative controls, human skin microbes would be filtered from the analysis, which would preclude the observation that many of the ISS surfaces were dominated by human-associated microbes. Was this filter applied before other analyses, or only for this analysis? If only for this analysis, the authors should report the values with and without the filter applied. If for all analyses, the authors should use a better approach for integrating negative controls, if they seem important to integrate. Perhaps this method would be useful: https://benjjneb.github.io/decontam/vignettes/decontam_intro.html

We are very thankful for this comment and appreciate discussion on proper handling of controls and false positives.

This filter was applied before all analyses and while we are aware of the issue that this conservative approach may also remove some RSVs which are present in the samples as well as the negative controls, we consider it to be the most secure way of avoiding false positives, especially for low biomass environments such as the ISS and clean rooms.

Analysis on RSV level, compared to OTU level, allows a more targeted identification of potential contaminating sequences and did not harm the overall outcome.

If there would be human contamination of the negative controls, as in the mentioned example, it could also not be excluded that the samples itself would bear contamination independent of the sampling and thus this sequences should be in our opinion removed (although we would like to stress that this obviously was not the case, since the filtered data were dominated by human associated taxa).

We are very experienced in analysing low biomass environments, and we always decided to follow the most conservative way of analysis, which means here that we remove all potential contaminants. Moreover, in low biomass samples, the probability of leakage between the samples (and into the negative control) is much lower.

Decontam has been tested and was found to be useful (and is used in our laboratory) for high biomass samples, such as human microbiome samples.

☹ Line 397: The rarefaction process is not clearly explained here. Usually this is performed to achieve an even number of sequences per sample, not RSVs per sample. The description seems to be wrong as in the previous sentence the authors say they detected differences in the number of RSVs, but then say that all samples were subsampled to have the same number of RSVs. I think this is just an error in the description provided here. If the samples were rarefied to 649 sequences per sample, that is a very low number that should be clearly justified. With the technologies used here, we would typically see somewhere on the order of tens of thousands of sequences per sample.

Thank you very much for this comment! True, this was a typo and is now corrected. Samples were rarefied to 649 READS for the Inverse Simpson's analysis (Fig. 3), which was the smallest amount of filtered reads obtained from sample A8. This low threshold was chosen to include all samples in this specific analysis. For all other analysis, TSS normalization was used, as described in the materials and methods. The full number of sequences derived is given in Supplementary Table S1, the RSV table.

In general, the ISS is a low biomass environment which often leads to a low read-yield, as for example also reported by Ichijo et al. 2016, even after a nested approach, or even no reads at all for some samples, or leading to whole genome amplification before shotgun sequencing as e.g. most recently described in Singh et al 2018 and Be et al. 2017.

Ichijo, T., Yamaguchi, N., Tanigaki, F., Shirakawa, M. and Nasu, M. (2016) 'Four-year bacterial monitoring in the International Space Station—Japanese Experiment Module “Kibo” with culture-independent approach', *npj Microgravity*, 2(1), p. 16007. doi: 10.1038/npjmgrav.2016.7.

Be, N. A., Avila-Herrera, A., Allen, J. E., Singh, N., Checinska Sielaff, A., Jaing, C. and Venkateswaran, K. (2017) 'Whole metagenome profiles of particulates collected from the International Space Station', *Microbiome*. *Microbiome*, 5(1), p. 81. doi: 10.1186/s40168-017-0292-4.

Singh, N. K., Wood, J. M., Karouia, F. and Venkateswaran, K. (2018) 'Succession and persistence of microbial communities and antimicrobial resistance genes associated with International Space Station environmental surfaces', *Microbiome*, 6(1), p. 204. doi: 10.1186/s40168-018-0585-2.

☹ Some important questions remain about the change in microbiome composition over time in this data. Is variability over time on ISS significantly greater than variability over time on the most similar ground-based indoor environment surfaces, or is this degree of variation to be expected? Was the degree of variation over time greater than or less than the variation across sites from the same sampling event?

As there was also a comment from the other reviewer on similar issues, we decided to rephrase the specific sentences, and are not using the word “dynamic” anymore; overall, the statements were toned down. Whether the variability over time in the ISS is different to other environments is very difficult to assess, as ALL ground-based environments are somewhat connected to the outer biosphere. We were amazed, that although there is not contact with nature, the microbiome aboard the ISS is not completely stable in composition.

However, we considered performing additional analysis on the change in microbiome composition over time (e.g. “rate of change” analysis, Qiime2), however, all possible analyses require a higher number of replicates, and also additional time points, which we cannot provide. So unfortunately, we cannot address these questions raised by the reviewer, although we would also like to answer this question.

☞ Line 479: "besides potentially food for Lactococcus)." it's not clear what the authors mean here.

Lactococcus is frequently found in everyday dairy products and used as a probiotic. So we assumed, it could be a signature from the astronauts' meals and supplements. As the statement was misleading, we rephrased the sentence and removed the food reference. (199 ff)

☞ Line 727-737: "The ISS is a source of novel microbial species" This is overstated - it is a big jump to claim that these are newly identified species. Please provide the sequences for these eleven organisms, and the results from BLASTing them against the NCBI databases.

Changed to "The ISS possibly harbours previously undescribed species" (384 ff). Supplementary table added including sequence-accession numbers and similarity according to EzBiocloud, which proves superior for taxonomic classification compared to NCBI Blast search. As all the listed isolates differ more than 2% in their 16S rRNA gene, they can at first glance be considered to be novel species, but of course this has to be checked in detailed characterization. The statement in the publication is much reduced ("might even qualify...").

Number of putative novel isolates corrected to seven, as we realised that the list of eleven also included clean room isolates which are not mentioned in this paragraph anymore due to character limitations.

Minor comments:

☞ Line 64: "loose microbial diversity" should be "lose microbial diversity"

Corrected.

☞ Line 297: definition of an RSV does not imply a "separate taxonomic classification". For example, organisms of the same species or strain can have some variation in their rRNA. This clause of the sentence should be removed.

Removed

Reviewer #2 (Remarks to the Author):

Mora et al. report on the International Space Station experiment EXTREMOPHILES. This work advances understanding of the ISS microbiome structure and function by, in a novel way, combining molecular (amplicon sequencing, metagenomic, whole genome sequence) and cultivation (enrichment, physiology, resistance screen, microscopy) analyses. The manuscript is timely, generally well written, puts forth important findings, and will be of interest to a broad audience (e.g., microbiology, space travel, public health, engineering). However, there are several issues that must be addressed, mostly to improve the introduction and discussion, prior to publication.

We are very thankful for your comments and appreciate your detailed questions.

Comments:

☞ The concept of the core ISS microbiome is intriguing, especially across the 5 ISS modules sampled. However, given the relatively short time frame of the study (May-July of 2017), the claim that the core is present over time becomes speculative. The discussion of this finding in context of previous work needs some improvement. Have all members of the core microbial community (all >50 genera) been detected in previous ISS expeditions with microbiome reports as well (e.g., Be et al. doi:10.1186/s40168-017-0292-4; Lang et al. doi:10.7717/peerj.4029)? This is important to support the claim of a core over time. Although you compare the “top 20” to the hospital microbiome study, it should also be compared to prior ISS expeditions. If the same core is not present, please explain why (perhaps human occupancy-related or possibly sequencing/primer bias) or change your presentation to describe a core microbiome within the time frame of the study. Would you expect “core” ISS microbial populations to change when a new crew arrives (could be an avenue for future research to present in the discussion)? Regarding all the above, please expand discussion following L871. Modify L 37-38, if needed, as well.

All core members were detected in previous studies. This information is now expanded in the discussion. (499 ff.)

☞ The introduction should be restructured to link some of the brief statements/paragraphs together. For example, human health, dependency on microbiome, and both being compromised in space seem go together. Then get into ISS microbiome concerns – including pathogens, resistance, materials damage. That can be followed with the ISS description as a sealed off vessel and little knowledge on microbiome dynamics in an enclosed built environment, let alone with spaceflight conditions. Piecing together the ideas will improve the flow of the story.

The introduction was improved, shortened and re-structured (space-flight goal, human health in space, risks through microorganisms, ISS microbiome and introduction of EXTREMOPHILES project). We hope the flow of the story was improved.

☞ L76-77: What were the problems? What happened? Be more specific

The problems were more clearly explained.(58 ff.)

☞ L96 and L101: Please don't start paragraphs with “publications focused on”. Instead just get to the key point and cite the source.

The paragraph was corrected accordingly.

☞ L109: Is “cosmonauts” necessary? Are they the same thing? Maybe change to “astronauts/cosmonauts”

Changed to “astronauts/cosmonauts” as requested. It is important to keep both names, as the Russian crew is referred to as cosmonauts.

☞ L112-116: Please restructure to form a stronger conclusion to the paragraph and overall background section. Try... “This may be attributable to environmental contamination limits (air, surfaces) having rarely been exceeded, at which times effective countermeasures had been implemented (24). Moreover, a genomics-based meta-analysis demonstrated that although pangenomes of *Bacillus* and *Staphylococcus* isolated from the ISS differed from Earth-based counterparts, these differences did not appear to be health-threatening (25). A comprehensive understanding of potential microbial adaptations to the ISS based on complementary genomic and cultivation approaches is needed to better understand potential human health implications (e.g., antibiotic resistance dissemination).”

These aspects were included in the introduction. (90 ff.)

☞ L118-123: Re-write section as the objectives of the study.
Section was adapted accordingly.

☞ L305: Re-name: “Microbial Community Network” or “Microbial Interaction Network”

Corrected.

☞ L343: Explicitly state a lack of sample replication (if that is true). It’s not just the number of samples taken, but also the number of replicates. Replicates are critical for microbiome studies and a lack may also explain drastic differences across location and time point.

Included (774 ff.). As stated in Table 1, each location was sampled only once per session. Replicates of the same location in the same session would also have been impractical, as the first wiping may already reduce the microbial load. Considering this, the astronaut was instructed to sample a rather large area of 1m² with each wipe. Sampled area size added to methods.

☞ L355-362: This information seems redundant with the methods. Remove.

Information reduced substantially. As Materials and Methods is now located behind the discussion, it might be helpful for the readers to quickly explain the concept of the sampling.

☞ Fig. 2: Panel A should say “RSV” not “OTU”. The caption should be more succinct and can begin with: “Temporal transitions in microbiome taxonomic diversity (same locations were sampled in Session A and Session B).”

We completely agree, done.

☞ L392-408 and throughout the text: I think “dynamics” is a little too broad a term to express differences in only two points, especially considering limited replicates. Just be specific and say the microbiome changed between time A and time B. Please make corrections throughout the text to be more tentative on how “dynamic” the microbiome is, and just describe that it is subject to change. Again, the discussion could use some serious expansion on the role of microbial transfers in maintaining the microbiome. Also, how is microbial survival in the built environment (not necessarily the ISS) known to vary (at least for the dominant taxa that you identified)? Look to survival studies. This has implications for microbiome structure dependency on time.

“Dynamics” rephrased throughout our results. The discussion was adapted, also following your comment above. However, we decided not to include the microbial survival aspect; the manuscript had to be substantially shortened to follow the requirements of the journal, and additional discussion could not be added. But we agree with the reviewer, that survivability is an important aspect.

☞ L452: Could be re-named to “Local factors shape microbiome composition”. Also, open the section describing how local factors may actually vary to give premise for why locations have different microbiomes (e.g., human traffic, human activity, water, cleanliness and cleaning frequency, materials present)

We re-named the chapter (187 ff.). Unfortunately, we do not have any information (beyond human traffic/activity) with respect to local water/humidity fluctuations, cleanliness and cleaning frequency, or the detailed material composition, as these were not measured or information was not provided. However, an additional statement was added to the chapter.

☞ L479: This statement may be over-reaching as some environmental strains are clearly present too (e.g., Bacillus, Pseudomonas). Be specific and say that the majority of strains from the sequencing-based analysis appear human-derived.

Corrected.

☞ Fig 5: Are these all significant? Please indicate which correlations are actually significant in the chart

Unfortunately, significance values cannot be provided, as for some locations less than two sample points are available.

☞ L524-525: Where is this data presented? If nowhere, please add a supp figure.

This information is now included, Supplementary Fig. S7

• L534: Supp Fig 7 needs larger font. Also, consider presenting as a simple heatmap.

Figure was improved (now Supplementary Fig. S8)

☞ L552: Fig. 7 needs a scale for node size

Font size increased, largest nodes include numbers now.

☞ L593, L597: need to add p-values
P-values added.

☞ L711: change “(relevant)” to “dominant” as they may still be relevant.
Corrected.

☞ L714: should be Fig. S9 not S6
Corrected.

☞ L751-2: change “was found” to “was likely linked to its” and remove “by”
Corrected.

☞ L760: did not adhere to
Corrected.

☞ L815-816: Split to a new paragraph. L815 will need a concluding sentence.
Split into a new paragraph, sentence adapted.

☞ L810-820: What statistical method was used to compare the functional attribute relative abundances – ANOVA, as described in Fig 12? If so, was an appropriate post-hoc p-value correction performed, such as Bonferroni or B-H False Disc Rate? Please implement a statistical test to remove potential false positives from the broad dataset in which abundance values are intrinsically dependent on all other values in the data set. Also, the metagenome function data needs to be made available in supplementary material, just like the RSV Taxa Table S1.

Statistical test is now explained (Figure legend), and additional tests were performed. Metagenomic functional (and taxonomic) data are now available (Supplementary data 2 and 3).

☞ L836: remove the parenthesis statement as this probably has to do with more human microbes from more frequent human occupancy
Corrected.

☞ L877: change “fluctuations” to “differences”. Fluctuations implies that it changes over time, but it is really only two time points.
Corrected.

☞ L882: Paragraph needs a stronger conclusion. Lead into future need for research at multiple points over a larger time series to better understand microbiome evolution, transmission to humans, and health implications over time.
Conclusion added. (L 508)

☞ L889: What about limitations and bias in sequencing efforts as well? Please indicate. Also, it is important to note that DNA detection does not mean viability
Included. (516 ff)

☞ L899: “In our study we confirmed” should be changed to “Our results support” or “Our results are in line with” Also, this is a run-on sentence. Please modify.
Done.

- L919: How realistic was this simulation in relevance to bacteria degrading materials on the ISS? The inoculum was incubated with liquid media and a material for ~3 months (L274)? Seems excessive. Are there any spaces aboard that are completely submerged, and if so, for how long (is biofouling really an issue or just biofilms)? Put your results in this context and just emphasize the necessity to keep moisture down to prevent biofilm growth and, in the events that biofilm forms, potential subsequent issues with material damage may arise.

NOMEX and EN AW 2219 alloy surfaces are not expected to be completely submerged in liquid for such a long timeframe, our experiment indicates what could happen in a worst case scenario. Chapter edited in this context. (550 ff)

- Supp Fig. S5: OTU should be changed to RSV
Corrected.

- Throughout the text, clarity in some of the figures would benefit from larger font size
Font increased where possible without destroying the figure layout: Fig.6,7,8

- ➡ Fig 12: Perform Tukey post-hoc test on each ANOVA to indicate which of the three are same or different
Done and information included.

Reviewers' comments:

Reviewer #1 (Remarks to the Author):

From rebuttal letter:

> Basically, the type of display for the NMDS in Fig.4 was allowing us to show better that the groups are not clearly separating from each other.

Making different choices about which methods to use in different parts of the same study solely based on which better supports your results suggests that you're selecting the most supportive results and not presenting data that doesn't support your conclusions. The authors should choose one ordination approach to use consistently throughout the manuscript, or present results from both methods in all places.

My comment about using years-old software for network analysis wasn't addressed. I still recommend using a more modern approach (and data from the previous studies could of course be reanalyzed with more modern approaches if the motivation is to do a pooled analysis). I therefore have the same comment as before, and leave it to the editors to decide if they feel this is important.

My comment about false positive handling being too simplistic wasn't addressed. I therefore still have the same comment as before. I think that it is important to at least present parallel information without this filter so readers can assess how much this filter impacts the results, since it is likely to have an impact on the results. The description of this as a "conservative approach" is debatable and subjective: reads that may represent real sequences observed on ISS surfaces are liberally being excluded from the samples being analyzed. Additionally, this statement "Decontam has been tested and was found to be useful (and is used in our laboratory) for high biomass samples, such as human microbiome samples" doesn't provide relevant information on this topic. Has it been applied to low biomass samples and found to not be useful? Why wasn't it applied here?

> The full number of sequences derived is given in Supplementary Table S1, the RSV table.

The rows in the table are species assignments, so this is more accurately described as a species or taxon table. The file itself contains a note that it is the RSV table, so that should probably be corrected.

I don't see this information in that table, but I computed it by summing the counts in each column. 649 is a very low value to use here - this is less than 7% of the total number of sequence reads that were collected in this study. I recommend that analyses performed with this data are also run at a higher number of sequences per sample and compared, so readers can assess if this result is consistent to one based on a lot more of the data that was collected.

Reviewer #2 (Remarks to the Author):

The authors have done a nice job revising the manuscript and updating the figures. I only have a few, very minor comments:

- Line 105: remove "were"
- Line 155: change "on" to "at"
- Line 200: insert "and" before "Neisseria"
- Line 428: Perhaps "human body sites" is more appropriate?

- Line 474: Using the transition "Accordingly" may be more appropriate here
- Line 535: Join paragraphs. Also, consider replacing "rather argue" with "propose"

Rebuttal letter

Dear reviewers,

We addressed all comments, as detailed below. Thank you very much for your support, and the time and effort you have spent to improve our manuscript! We appreciate all comments and the raised discussion!

Reviewer #1 (Remarks to the Author):

** From rebuttal letter:*

Basically, the type of display for the NMDS in Fig.4 was allowing us to show better that the groups are not clearly separating from each other.

Making different choices about which methods to use in different parts of the same study solely based on which better supports your results suggests that you're selecting the most supportive results and not presenting data that doesn't support your conclusions. The authors should choose one ordination approach to use consistently throughout the manuscript, or present results from both methods in all places.

We are sorry for our misleading response. The NMDS of Fig. 3 was now replaced by a PCoA plot. Our conclusions are also supported by the PCoA plot (groups are not clearly separating from each other).

** My comment about using years-old software for network analysis wasn't addressed. I still recommend using a more modern approach (and data from the previous studies could of course be reanalyzed with more modern approaches if the motivation is to do a pooled analysis). I therefore have the same comment as before, and leave it to the editors to decide if they feel this is important.*

As the editor did not request changes on the network analysis, the network was not changed. We appreciate your comment and will for sure consider in subsequent studies, when detailed microbial interactions are in the focus of the analysis.

** My comment about false positive handling being too simplistic wasn't addressed. I therefore still have the same comment as before. I think that it is important to at least present parallel information without this filter so readers can assess how much this filter impacts the results, since it is likely to have an impact on the results. The description of this as a "conservative approach" is debatable and subjective: reads that may represent real sequences observed on ISS surfaces are liberally being excluded from the samples being analyzed. Additionally, this statement "Decontam has been tested and was found to be useful (and is used in our laboratory) for high biomass samples, such as human microbiome samples" doesn't provide relevant information on this topic. Has it been applied to low biomass samples and found to not be useful? Why wasn't it applied here?*

Parallel information (based on a decontam dataset) is now provided, see Supplementary Note 3 and Supplementary Data 5 (all decontam RSV tables). In Materials and Methods we added a reference to both additional files. In Supplementary Note 3, we wrote: "In addition to the approach used in the main body of the manuscript, which presents results based on data which were completely cleaned

by all RSVs detected in the negative controls, we determined contaminating RSVs also via decontam (Davis et al., 2018). Decontam identified 68 RSVs which were subsequently removed from the RSV table (Supplementary Data 5).

However, the cleaned RSV table still contained typical „kitome“ microbial signatures which were present in both, samples and negative controls (Salter et al., 2014; e.g. *Acinetobacter*, *Alcaligenes*, *Bacillus*, *Bradyrhizobium*, *Herbaspirillum*, *Mesorhizobium*, *Methylobacterium*, *Microbacterium*, *Novosphingobium*, *Pseudomonas*, *Ralstonia*, *Sphingomonas*, *Stenotrophomonas* and *Xanthomonas*). Thus we decided to follow a very conservative approach for the main manuscript. For the sake of completeness, the analysis of the decontam- dataset is presented herein.”

In Supplementary Note 3 we provide a decontam dataset-based version of Fig. 1, Fig. 3, Supplementary Fig. 3, Supplementary Fig. 5, Supplementary Fig. 10.

The decontam-based analyses, largely confirmed our findings, such as:

- The most dominant phyla detected are Proteobacteria, Firmicutes, Actinobacteria and Bacteroidetes.
- Instead of *Streptococcus*, *Ralstonia* was identified to the predominant signature, which might, however, represent a kitome representative (see Salter et al.).
- The comparison of session A and B confirmed a significant increase in diversity, and a shift of the microbial profile. In addition, the increase of gut-associated microorganisms was confirmed (*Escherichia/Shigella*, *Pseudobutyrvibrio*, *Ruminococcus*, *Bacteroides* etc.)
- The microbiome profile did still not group according to sample category (Supplementary Note Fig. 2), and archaeal signatures remained indicative of locations.
- Clean room samples grouped separated from ISS samples.

To conclude, the additional analysis based on decontam did not change our major messages. Both analysis results and raw files are now available for the reader.

We appreciate the discussion with the reviewer, and we hope that we satisfactorily addressed all open questions.

* The full number of sequences derived is given in Supplementary Table S1, the RSV table.

The rows in the table are species assignments, so this is more accurately described as a species or taxon table. The file itself contains a note that it is the RSV table, so that should probably be corrected.

It is the RSV table, and the first row shows the classification of the RSV's down to species level (wherever possible). Information on columns was now added, as well as the number of the respective RSV.

* I don't see this information in that table, but I computed it by summing the counts in each column. 649 is a very low value to use here - this is less than 7% of the total number of sequence reads that were collected in this study. I recommend that analyses performed with this data are also run at a higher number of sequences per sample and compared, so readers can assess if this result is

consistent to one based on a lot more of the data that was collected.

It shall be mentioned that only the diversity analysis requires the normalization to a certain number of reads, all other analyses are not affected by normalization at a low read count.

As requested, we repeated the diversity analysis at increased sequence quantities. All additional diagraphs are given below.

We subsequently removed samples with the lowest read count to reach normalization at higher sequence counts. However, even when we normalized at 3201 sequences (by removing samples A8, C1, A7, A5, A9, A3), the outcome did not change: the diversity was found not to be significantly different to other area categories, and personal areas always revealed the highest diversity.

As we would like to keep all our samples for the analyses, we propose to keep the plot as shown in the current version. If the editor decides to add the additional analyses, we are happy to do so.

Figure 1: Rarefied at 1002, removed sample A8

Figure 2: Rarefied at 1040, removed sample A8 and C1

Figure 3: Rarefied at 1273, removed samples A8, C1, A7

Figure 4: rarefied at 1296, removed samples A8, C1, A7, A5

Figure 5: rarefied at 1783, removed samples A8, C1, A7, A5, A9

Figure 6: rarefied at 3201, removed samples A8, C1, A7, A5, A9, A3

Reviewer #2 (Remarks to the Author):

The authors have done a nice job revising the manuscript and updating the figures. I only have a few, very minor comments:

Thank you, all minor comments were addressed.

- Line 105: remove “were”

Corrected.

- Line 155: change “on” to “at”

Corrected.

- Line 200: insert “and” before “Neisseria”

Corrected.

- Line 428: Perhaps “human body sites” is more appropriate?

Corrected.

- Line 474: Using the transition “Accordingly” may be more appropriate here

Corrected.

- Line 535: Join paragraphs. Also, consider replacing “rather argue” with “propose”

Corrected.

REVIEWERS' COMMENTS:

Reviewer #1 (Remarks to the Author):

This revision addresses the concerns that I brought up which were shared by the editor, and the responses to those requests are technically sound.

My final recommendation is that you mention in the text that parallel analyses were performed at multiple even sampling depths, and that results were consistent. You can indicate "data not shown".

In Figure 3, note that the shape/color combinations in the legend don't align with the shape/color combinations in panels a or b.

We are very grateful for the support by the reviewer and the critical discussion.
We corrected all issues raised.

REVIEWERS' COMMENTS:

Reviewer #1 (Remarks to the Author):

This revision addresses the concerns that I brought up which were shared by the editor, and the responses to those requests are technically sound.

My final recommendation is that you mention in the text that parallel analyses were performed at multiple even sampling depths, and that results were consistent. You can indicate "data not shown".

This information is now included. L 195-197.

In Figure 3, note that the shape/color combinations in the legend don't align with the shape/color combinations in panels a or b.

Fig. 3 was corrected.